# Prethermalization in coupled one-dimensional quantum gases

Maciej Łebek[1]⋆, Miłosz Panfil[1] and Robert M. Konik[2]

**1** Faculty of Physics, University of Warsaw, Pasteura 5, 02-093 Warsaw, Poland
**2** Condensed Matter Physics and Materials Science Division,
Brookhaven National Laboratory, Upton, NY 11973, USA

⋆ maciej.lebek@fuw.edu.pl

## Abstract

We consider the problem of the development of steady states in one-dimensional Bose gas tubes that are weakly coupled to one another through a density-density interaction. We analyze this development through a Boltzmann collision integral approach. We argue that when the leading order of the collision integral, where single particle-hole excitations are created in individual gases, is dominant, the state of the gas evolves first to a non-thermal fixed point, i.e. a prethermalization plateau. This order is dominant when a pair of tubes are inequivalent with, say, different temperatures or different effective interaction parameters, $\gamma$. When both tubes are in the strongly interacting regime we additionally characterize this non-thermal prethermalization plateau by constructing the quasi-conserved quantities that control the existence of this plateau as well as the associated generalized Gibbs ensemble.

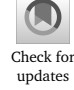

# 1   Introduction

How to understand the dynamics of a quantum system after the injection of energy is a paramount problem of modern many-body physics [1–3]. The injection of energy typically leads a system to arrive at a new steady state that is thermal in nature. The process by which this occurs in closed quantum systems is governed by the eigenstate thermalization hypothesis (ETH) [4–8]. This hypothesis asserts that quantum eigenstates of similar energy behave similarly with respect to all local observables, at least in the thermodynamic limit. However this hypothesis is not inviolate and there are models where ETH does not hold. One such class of ETH violating systems that has been extensively studied in the past few years support what are known as quantum scar states [9–12]. Quantum scar states have expectation values relative to some local observable that differs significantly from most other system eigenstates at the same energy. When a system is then in a scar state, it will experience atypical time dynamics relative to that observable and will not thermalize in the sense of ETH.

Another class of ETH violating systems are integrable models. Integrable models are characterized by an infinite set of conservation laws. The presence of these conservation laws means that their steady states are characterized by more complicated thermodynamic ensembles than Gibbs that take into account the presence of conserved charges beyond energy [8,13] Integrable models are ubiquitous in one-spatial dimension. Such models describe the dynamics of cold atomic Bose gases via the Lieb-Liniger model [14,15], XXZ spin chains via the Heisenberg spin model [16], and itinerant interacting electron physics via the Hubbard model [17,18] to name but a few.

Integrable models are to a certain degree platonic ideals. They only exist approximately in nature. Experimental realizations of one-dimensional Bose gases modelled by a Lieb-Linger (LL) model involve arrays of one-dimensional tubes of the gas which see both intra- and intertube interactions beyond the interactions present in the LL Hamiltonian [19–21]. Similarly, quantum material realizations of one-dimensional spin chains are always in fact one-dimensional atomic chains embedded in a three dimensional matrix with interchain interactions, perhaps small but nonetheless, present [22,23]. Finally, also material realizations of low-dimensional Hubbard models typically require the reduction of complicated multi-orbital physics down to effective one-band models [24–27].

The interactions that go beyond those present in integrable Hamiltonians will almost always break integrability. While an integrable model will not thermalize, a quasi-integrable model will (apart from cases when quantum scars are present). However it then becomes a

question of timescales. Rather than a non-thermal steady state persisting for all time, it instead picks up a finite, but perhaps long, lifetime. This persistent non-thermal steady state is known as a prethermalization plateau [28,29]. This prethermalization plateau can usually be characterized by conserved quantities other than energy. However these conserved quantities are not necessarily the same as in the parent, unperturbed, integrable model.

It is the aim of this paper to present a scenario relevant to experiments in one-dimensional cold atomic Bose gases where long-live prethermalization plateaus occur. In typical realizations of one-dimensional Bose gases, the atoms in the gas are placed in an laser-induced array of quasi-one-dimensional transversal confining potentials, i.e. the gas forms of matrix of tubes [19–21]. The atoms in each tube are typically confined to the lowest transverse level of the tube but are free to move laterally along the tube. However the gas in each tube is not entirely independent of its neighbours in the array. Typically intertube density-density interactions are present. Such interactions in gases with dipolar couplings are tunable and can even be made to be of the same order as intratube interactions [19].

In Ref. [30] a framework for analyzing the thermalizing effects of intertube interactions was developed in the form of a Boltzmann collision integral. In Ref. [30], the tubes of gas were considered to be identical. A key feature of this work was then that the thermalizing interactions involve processes of the simultaneous creation of three particle-hole pairs in any two tubes: one pair in one tube and two pairs in a second tube, so-called (2,1) or (1,2) processes. Processes involving the production of a single pair in two identical tubes, (1,1) processes, led to no change in the state of the tubes, a simple consequence of energy-momentum conservation.

If however the two tubes are inequivalent, either by virtue of having a different density of atoms or a different effective interaction parameter, (1,1) processes are dominant at shorter time scales and do lead to an evolution in the state of the gas in the tube. However here the end point of this evolution is athermal. Because we cannot ignore (1,2)/(2,1) processes, this athermal state arrived at by the creation of single particle-hole pairs is temporary – it is in fact a prethermalization plateau. Nonetheless it is long-lived if the intertube interactions, which set the scale of the evolution, are weak. It is the aim of this paper to describe in detail the non-equilibrium evolution coming from tube heterogeneity and the resulting athermal state.

While the focus of this analysis takes place in the context of cold atomic systems, the analysis has elements of universality. A very similar set of considerations, for example, would apply to thermalizing $S_z S_z$ interchain interactions in spin chain materials. The universality arises because in the thermodynamic limit the matrix elements governing the creation of particle hole pairs (and their spin chain equivalents) have the same low energy form [31,32].

The paper is organized as follows. In Section 2 we outline the basics of the Lieb-Liniger model including how to describe its thermodynamic states and its conserved quantities. In the next section, Section 3, we present the Boltzmann collision framework by which we analyze the dynamics of two tubes of gas due to intertube interactions. As part of this we present a general analysis of what constitutes a steady state in this framework and what are its conservation laws. We do this in complete generality considering all orders of particle-hole creation (i.e. (n,m)) processes. As part of this, we show that only energy, momentum, and particle number survive as conserved quantities and the resulting steady state is thermal.

In section 4 we perform the same analysis but only consider (1,1) processes. Here we show that with only (1,1) processes taken into account, there exist stationary states there are athermal. We then demonstrate by numerically solving the problem that indeed the system does not thermalize when only (1,1) processes are operative. In Section 5 we offer an explanation for the athermality by showing that the (1,1) dynamics has extra conserved charges beyond the energy, momentum and particle number. The new conserved charges are combination of the ultra-local charges of unperturbed Lieb-Liniger model from both tubes. We construct these charges for systems with strong intra-tube interactions, dubbed deformed Tonks-Girardeau

gas. We define it precisely in Section 5. We also formulate the Generalized Gibbs Ensemble for the coupled system essentially extending its applicability to perturbed integrable models. Finally, in Section 6, we provide a summary and concluding remarks.

More technical aspects of our work are relegated to the Appendices. Among them, in Appendix E, we demonstrate that the construction of the charges from the deformed Tonks-Girardeau gases does not extend to the full Lieb-Liniger model at arbitrary interaction strengths. This leaves open a question about the precise characterisation of the mechanism by which the system does not thermalize in the latter case.

## 2 Summary of the Lieb-Liniger model

We start with reviewing the relevant aspects of the Lieb-Liniger (LL) model [14, 15], which corresponds to a single gas tube. The LL model describes $N$ bosons on a ring with length $L$, interacting with a repulsive contact potential. The Hamiltonian of the system reads

$$\hat{H}_{LL} = \int_0^L dx \Big( -\hat{\psi}^\dagger(x)\partial_x^2\hat{\psi}(x) + c\hat{\psi}^\dagger(x)\hat{\psi}^\dagger(x)\hat{\psi}(x)\hat{\psi}(x) \Big), \tag{1}$$

where $c \geq 0$ is the coupling constant and $\hat{\psi}(x)$ is the canonical bosonic field operator. The model is exactly solvable with the standard Bethe ansatz technique [33–35]. Eigenstates of (1) are parametrized by $N$ real numbers, called quasimomenta, which are determined from the Bethe equations. In the thermodynamic limit $N, L \to \infty$ with $N/L = \text{const}$, the system is parametrized by the density of particles, $\rho_p(\lambda)$, and the total density of states, $\rho_t(\lambda)$, with $\lambda$ the quasi-momentum or rapidity. These functions are related through the following integral equation [36],

$$\rho_t(\lambda) = \frac{1}{2\pi} + \int d\lambda' \, T(\lambda - \lambda')\rho_p(\lambda'), \tag{2}$$

with the kernel given by function, $T(\lambda)$:

$$T(\lambda) = \frac{c}{\pi}\frac{1}{c^2 + \lambda^2}. \tag{3}$$

In addition to $\rho_t$ and $\rho_p$, we define the density of holes, $\rho_h$, as

$$\rho_h(\lambda) = \rho_t(\lambda) - \rho_p(\lambda). \tag{4}$$

It is sometimes convenient to express thermodynamic quantities in terms of $\rho_h$ rather than $\rho_p$.

The LL model is integrable and so is characterized by an infinite number of conserved charges, $\hat{I}_n$, that commute with the Hamiltonian (1) and have the property

$$\big[\hat{I}_n, \hat{I}_m\big] = 0, \qquad n, m = 0, 1, 2, \dots \tag{5}$$

There are multiple bases of these charges, both ultra-local [37] and semi-local [38–40]. In this work we will focus on the ultra-local representation despite their less than stellar behaviour in the UV [41]. Expectation values of the ultra-local conserved charges on thermodynamic states are functionals of the distribution, $\rho_p(\lambda)$, and take the particularly simple form

$$\langle\hat{I}_n\rangle = L \int d\lambda \, \rho_p(\lambda)\lambda^n. \tag{6}$$

The first operators in the sequence, $\hat{I}_n$, correspond to particle number, total momentum and total energy, $n = 0, 1, 2$, respectively:

$$N = L \int \mathrm{d}\lambda \rho_p(\lambda), \quad P = L \int \mathrm{d}\lambda\, \lambda \rho_p(\lambda), \quad E = L \int \mathrm{d}\lambda\, \lambda^2 \rho_p(\lambda). \tag{7}$$

The presence of conserved charges $n > 2$ beyond these basic three means that the model can support non-thermal states.

We now turn to characterizing the thermodynamics and elementary excitations of the LL model. The entropy of a thermodynamic state of the gas is given by the Yang-Yang formula [36]

$$S = L \int \mathrm{d}\lambda \Big[ \big( \rho_p(\lambda) + \rho_h(\lambda) \big) \log \big( \rho_p(\lambda) + \rho_h(\lambda) \big) - \rho_p(\lambda) \log \rho_p(\lambda) - \rho_h(\lambda) \log \rho_h(\lambda) \Big]. \tag{8}$$

To describe the dynamics in the system we also need information about its elementary excitations. The coupling between the tubes that we consider, conserves number of particles in each tube. Therefore the relevant excitations takes form of particle-hole (ph) excitations in the system [15, 42]. The excitations are labelled by two quasimomenta $p$ and $h$. The momentum carried by an excitation is $k = k(p) - k(h)$ and the corresponding energy is $\omega = \omega(p) - \omega(h)$. Here, the functions $k(p)$ and $\omega(p)$ are the so-called dressed momentum and energy. These are state-dependent functions and read

$$k(\lambda) = \lambda - \int \mathrm{d}\mu\, n(\mu) F(\mu|\lambda), \tag{9a}$$

$$\omega(\lambda) = \lambda^2 - 2 \int \mathrm{d}\mu\, \mu n(\mu) F(\mu|\lambda), \tag{9b}$$

where $n(\lambda) = \rho_p(\lambda)/\rho_t(\lambda)$ is the occupation function and $F(\mu|\lambda)$ is the backflow function [42] satisfying the following integral equation,

$$F(\lambda|\mu) = \frac{\theta(\lambda - \mu)}{2\pi} + \int \mathrm{d}\lambda'\, T(\lambda - \lambda') n(\lambda') F(\lambda'|\mu), \tag{10}$$

with $\theta(\lambda) = 2 \arctan(\lambda/c)$.

For later use, we introduce notation for two dressing operations, following the notation of [43]. The first one, denoted Dr, is defined as

$$f_{Dr}(\lambda) = f(\lambda) - \int \mathrm{d}\mu\, n(\mu) F(\mu|\lambda) n(\mu) f'(\mu). \tag{11}$$

Therefore $k(\lambda) = (\lambda)_{Dr}$ and $\omega(\lambda) = (\lambda^2)_{Dr}$. The second one, denoted dr, is defined as

$$f_{dr}(\lambda) = f(\lambda) + \int \mathrm{d}\mu\, T(\mu, \lambda) n(\mu) f_{dr}(\mu). \tag{12}$$

The two dressing operations are related however through

$$(f_{\mathrm{Dr}})' = (f')_{\mathrm{dr}}. \tag{13}$$

## 3 Coupled Lieb-Liniger gases

In this section, we present a framework to study the dynamics of two coupled LL gases. The system is governed by the following Hamiltonian,

$$\hat{H} = \hat{H}_{LL,1} + \hat{H}_{LL,2} + \int \mathrm{d}x_1 \mathrm{d}x_2\, A(x_1 - x_2) \hat{\rho}_1(x_1) \hat{\rho}_2(x_2). \tag{14}$$

The Hamiltonians $\hat{H}_{LL,i}$ describe LL gases with couplings $c_i$. The two gases are coupled by a long-range interaction potential $A(x_1 - x_2)$. By $\hat{\rho}_i(x) = \hat{\psi}_i^\dagger(x)\hat{\psi}_i(x)$, we denote the density operator in the $i$-th gas. The density-density interaction term $\delta\hat{H}$ breaks the integrability of the system $\hat{H} = \hat{H}_{\text{int}} + \delta\hat{H}$, where $\hat{H}_{\text{int}} = \hat{H}_{LL,1} + \hat{H}_{LL,2}$ is integrable. To describe its consequences on the gases' dynamics, we turn to an effective theory of relaxation developed in [30, 44]. Specifically in [30] the problem of the thermalization of two equivalent gases initialized in a non-thermal state was analyzed in quantitative detail. On the level of Fermi's golden rule approximation, the dynamics may be formulated in terms of Boltzmann-like equations governing the evolution of the rapidity distributions:

$$
\begin{aligned}
\partial_t \rho_{p,1}(\lambda) &= \tau^{-1} Q[\rho_{p,1}, \rho_{p,2}](\lambda)\,, \\
\partial_t \rho_{p,2}(\lambda) &= \tau^{-1} Q[\rho_{p,2}, \rho_{p,1}](\lambda)\,,
\end{aligned}
\tag{15}
$$

where $\tau$ is the characteristic time scale for the evolution and $Q[\rho_{p,1}; \rho_{p,2}, \lambda]$ is the dressed scattering integral defined through

$$
\begin{aligned}
Q[\rho_{p,1}, \rho_{p,2}] &= \mathbb{F}_1 \cdot Q_0\left[\rho_{p,1}, \rho_{p,2}\right]\,, \\
Q[\rho_{p,2}, \rho_{p,1}] &= \mathbb{F}_2 \cdot Q_0\left[\rho_{p,2}, \rho_{p,1}\right]\,,
\end{aligned}
\tag{16}
$$

where $Q_0[\rho_{p,1}, \rho_{p,2}]$ is bare scattering integral defined below in Eq. (22) and $\mathbb{F}_i(\lambda, \mu)$ reads

$$
\mathbb{F}_i(\lambda, \mu) = \delta(\lambda - \mu) + \partial_\lambda(n_i(\lambda)F_i(\lambda|\mu))\,,
\tag{17}
$$

and implements the dressing operation, Dr, defined in Eqn. (11). We also define

$$
(\mathbb{F} \cdot f)(\lambda) = \int d\mu\, \mathbb{F}(\lambda, \mu) f(\mu)\,.
\tag{18}
$$

The first term in Eqn. (17) represents a direct scattering process while the second term gives the so-called backflow, i.e., the effect, due to interactions, that the creation of a particle-hole excitation has on the distribution of the remaining particles and holes in the gas.

The time scale, $\tau$, appearing in (15) is determined by the parameters of the model [30]

$$
\tau^{-1} = \frac{2A_0^2 m}{\hbar^3}\,,
\tag{19}
$$

where $m$ is mass of the particles and $A_0$ sets the strength of the inter-tube interaction,

$$
A(x) = A_0 \frac{a(x/x_r)}{x_r}\,, \qquad \int dx \frac{a(x/x_r)}{x_r} = 1\,.
\tag{20}
$$

Here $x_r$ is the characteristic range of the potential. We will mostly work in momentum space, and so we define the Fourier transform of the normalized part:

$$
\tilde{A}(k) = \int dx\, e^{ikx} \frac{a(x/x_r)}{x_r} = \tilde{a}(k/k_r)\,, \qquad \tilde{a}(k) = \int dx\, e^{ikx} a(x)\,,
\tag{21}
$$

where $k_r = x_r^{-1}$ is the characteristic momentum exchanged between the tubes as a result of the coupling between them.[1]

---

[1] For all the simulations in the paper, we use the Gaussian potential $a(x) = 1/(\sqrt{\pi})\exp(-x^2)$ with $x_r = 11$.

### 3.1 (n,m) ph processes

The bare collision integral $Q_0[\rho_{p,1};\rho_{p,2}]$ contains contributions from processes involving $n$ particle-holes (ph) excitations in the first tube and $m$ ph excitations in the second tube:

$$Q_0[\rho_{p,1},\rho_{p,2}] = \sum_{n,m} Q_0^{(n,m)}[\rho_{p,1},\rho_{p,2}]. \tag{22}$$

The $(n,m)$ contribution takes the following form [30],

$$Q_0^{(n,m)}[\rho_{p,1},\rho_{p,2}](\lambda) = \int d\mathbf{p}^n d\mathbf{h}^n \sum_{i=1}^{n} \delta(\lambda - p_i)\tilde{A}^2(k_1(\mathbf{p},\mathbf{h})) |F_1(\mathbf{p},\mathbf{h})|^2$$
$$\times \left( J_1^n(\mathbf{p},\mathbf{h})S_2^m(-k_1(\mathbf{p},\mathbf{h}),-\omega_1(\mathbf{p},\mathbf{h})) - (\mathbf{h} \leftrightarrow \mathbf{p}) \right), \tag{23}$$

where $\mathbf{p} = \{p_i\}_{i=1}^n$ and $\mathbf{h} = \{h_i\}_{i=1}^n$ denote rapidities of particles and holes, respectively. Here we have introduced the notation

$$d\mathbf{p}^n d\mathbf{h}^n = \frac{1}{(n!)^2} \prod_{i=1}^{n} dp_i dh_i, \qquad J_1^n(\mathbf{p},\mathbf{h}) = \prod_{i=1}^{n} \rho_{p,1}(h_i)\rho_{h,1}(p_i). \tag{24}$$

Moreover by $F(\mathbf{p},\mathbf{h})$ we denote the density operator form factor [45–48]

$$F(\mathbf{p},\mathbf{h}) = \langle \rho_p|\hat{\rho}(0)|\rho_p;\mathbf{p},\mathbf{h}\rangle, \tag{25}$$

where the state $|\rho_p;\mathbf{p},\mathbf{h}\rangle$ corresponds to the thermodynamic state $|\rho_p\rangle$ with ph excitations $\mathbf{p}$ and $\mathbf{h}$ on top of it. Finally, $S^m(k,\omega)$ is the $m$-particle contribution to the dynamic structure factor [46]. In Eqn. (23), we denote the total momentum and energy carried by excitations $\mathbf{h} \to \mathbf{p}$ in the first tube by $k_1(\mathbf{p},\mathbf{h})$ and $\omega_1(\mathbf{p},\mathbf{h})$:

$$k_1(\mathbf{p},\mathbf{h}) = \sum_{i=1}^{n}(k_1(p_i) - k_1(h_i)), \qquad \omega_1(\mathbf{p},\mathbf{h}) = \sum_{i=1}^{n}(\omega_1(p_i) - \omega_1(h_i)). \tag{26}$$

For completeness, we also write down the bare collision integral for the second tube,

$$Q_0^{(n,m)}[\rho_{p,2},\rho_{p,1}](\lambda) = \int d\mathbf{p}^n d\mathbf{h}^n \sum_{i=1}^{n} \delta(\lambda - p_i)\tilde{A}^2(k_2(\mathbf{p},\mathbf{h})) |F_2(\mathbf{p},\mathbf{h})|^2$$
$$\times \left( J_2^n(\mathbf{p},\mathbf{h})S_1^m(-k_2(\mathbf{p},\mathbf{h}),-\omega_2(\mathbf{p},\mathbf{h})) - (\mathbf{h} \leftrightarrow \mathbf{p}) \right), \tag{27}$$

which amounts to a permutation of tube indices.

The form of the collision integral presented in Eqn. (23) is not the most convenient for our purposes. We thus derive now an alternative representation of it, expressing the dynamical structure factor contributions in terms of quasi-particle densities. The $m$-particle contribution to the dynamic structure factor reads

$$S^m(k,\omega) = (2\pi)^2 \int d\mathbf{p}^m d\mathbf{p}^m \prod_{i=1}^{m} \rho_p(h_i)\rho_h(p_i)|F(\mathbf{p},\mathbf{h})|^2 \delta\big(k - k(\mathbf{p},\mathbf{h})\big)\delta\big(\omega - \omega(\mathbf{p},\mathbf{h})\big), \tag{28}$$

with the understanding the integrals require, in general, regulation [48]. Plugging the above representation into (23) leads to the desired formulation:

$$Q_0^{(n,m)}[\rho_{p,1},\rho_{p,2}](\lambda) = (2\pi)^2 \int d\mathbf{p}_1^n d\mathbf{h}_1^n d\mathbf{p}_2^m d\mathbf{h}_2^m \sum_{j=1}^{n} \delta(\lambda - p_{j,1})\tilde{A}^2\big(k_1(\mathbf{p}_1,\mathbf{h}_1)\big) |F_1(\mathbf{p}_1,\mathbf{h}_1)|^2$$
$$\times |F_2(\mathbf{p}_2,\mathbf{h}_2)|^2\, \delta\big(k_1(\mathbf{p}_1,\mathbf{h}_1) + k_2(\mathbf{p}_2,\mathbf{h}_2)\big)\delta\big(\omega_1(\mathbf{p}_1,\mathbf{h}_1) + \omega_2(\mathbf{p}_2,\mathbf{h}_2)\big)$$
$$\times \left( J_1^n(\mathbf{p}_1,\mathbf{h}_1)J_2^m(\mathbf{p}_2,\mathbf{h}_2) - (\mathbf{h} \leftrightarrow \mathbf{p}) \right), \tag{29}$$

and similarly for the second tube,

$$
\begin{aligned}
Q_0^{(n,m)}[\rho_{p,2},\rho_{p,1}](\lambda) = (2\pi)^2 \int \, & d\mathbf{p}_1^m d\mathbf{h}_1^m d\mathbf{p}_2^n d\mathbf{h}_2^n \sum_{j=1}^n \delta(\lambda - p_{j,2}) \tilde{A}^2\big(k_2(\mathbf{p}_2,\mathbf{h}_2)\big) |F_1(\mathbf{p}_1,\mathbf{h}_1)|^2 \\
& \times |F_2(\mathbf{p}_2,\mathbf{h}_2)|^2 \, \delta\big(k_1(\mathbf{p}_1,\mathbf{h}_1) + k_2(\mathbf{p}_2,\mathbf{h}_2)\big) \delta\big(\omega_1(\mathbf{p}_1,\mathbf{h}_1) + \omega_2(\mathbf{p}_2,\mathbf{h}_2)\big) \\
& \times \Big( J_1^m(\mathbf{p}_1,\mathbf{h}_1) J_2^n(\mathbf{p}_2,\mathbf{h}_2) - (\mathbf{h} \leftrightarrow \mathbf{p}) \Big).
\end{aligned}
\tag{30}
$$

This representation is the most explicit. It is convenient for the numerical evaluation of the collision integral and for analyzing its small momentum limit. From the particle-hole symmetry of density operator form factors [46] and Eqn. (29), it is easy to find yet another representation,

$$
\begin{aligned}
Q_0^{(n,m)}[\rho_{p,1},\rho_{p,2}](\lambda) = (2\pi)^2 \int \, & d\mathbf{p}_1^n d\mathbf{h}_1^n d\mathbf{p}_2^m d\mathbf{h}_2^m \tilde{A}^2\big(k_1(\mathbf{p}_1,\mathbf{h}_1)\big) |F_1(\mathbf{p}_1,\mathbf{h}_1)|^2 |F_2(\mathbf{p}_2,\mathbf{h}_2)|^2 \\
& \times \delta\big(k_1(\mathbf{p}_1,\mathbf{h}_1) + k_2(\mathbf{p}_2,\mathbf{h}_2)\big) \delta\big(\omega_1(\mathbf{p}_1,\mathbf{h}_1) + \omega_2(\mathbf{p}_2,\mathbf{h}_2)\big) J_1^n(\mathbf{p}_1,\mathbf{h}_1) \\
& \times J_2^m(\mathbf{p}_2,\mathbf{h}_2) \left( \sum_{i=1}^n \delta(\lambda - p_{i,1}) - \sum_{i=1}^n \delta(\lambda - h_{i,1}) \right).
\end{aligned}
\tag{31}
$$

Obviously, analogous formula for $Q_0^{(n,m)}[\rho_{p,2},\rho_{p,1}](\lambda)$ may be easily derived as well. This representation is especially convenient for our discussion of the conservation laws in Section 3.3 below.

Crucially the $(n,m)$ ph processes possess a hierarchy of importance. For the situations considered here the most relevant contribution is due to $(1,1)$ scattering. Processes involving more ph pairs contribute much less significantly. In particular, let us note that in the extreme limit of infinite interactions in both tubes $c_{1,2} = \infty$ we have $Q^{(n,m)} = 0$ for $m, n > 1$ and only the $(1,1)$ dynamics contributes. This can be seen on the level of the higher ph density form factors which go as $1/c^{n+m-2}$ and so are zero when more than one ph pair is considered [46] in the infinite interaction limit. The hierarchy between different contributions is also clearly visible in the small momentum expansion of scattering integrals, which is explained in detail in the Appendix A. Assuming that the dynamics is driven mainly by the processes involving small momentum transfer between the tubes, we show there that

$$
Q_0^{(n,m)} \sim k_r^{n+m} \times \left(1 + \mathcal{O}\left(k_r^2\right)\right).
\tag{32}
$$

Noting that $k_r$ is a small parameter (the case of long-range interactions) we clearly see that the lowest processes are the most important ones. Furthermore because the subleading corrections involve $k_r^2$, the hierarchy of the importance is as following:

1. The leading order $(1,1)$ processes going as $k_r^2$.

2. The leading $(2,1)$ and $(1,2)$ processes, of order $k_r^3$.

3. The subleading $(1,1)$ processes and the leading order of higher processes like $(3,1)$ or $(2,2)$, all of order $k_r^4$.

As we have observed in [30] the processes $(2,1)$ and $(1,2)$ thermalize the system. There we considered the situation of identical states in both tubes, and in such a case, the scattering integral for $(1,1)$ processes vanishes. But in the general case, the physics at short timescales is determined by the leading $(1,1)$ processes. We analyze these in Section 4. In the remainder of this section, we discuss the properties of the dynamics generated by (15) for arbitrary processes. We address specifically the character of stationary states and the existence of conserved charges.

## 3.2 Stationary states

In this subsection we look for stationary states satisfying

$$Q[\rho_{p,1}, \rho_{p,2}](\lambda) = Q[\rho_{p,2}, \rho_{p,1}](\lambda) = 0. \tag{33}$$

First, we note that when analyzing the equation above, we may simplify the problem and focus on the bare collision integral $Q_0$. This is because the invertible operation of dressing is linear and the vanishing of $Q_0$ implies the vanishing of $Q$. Let us start with the time evolution of the first tube. The bare scattering integral can vanish as a whole or because the integrand is identically zero. The second condition is stronger and implies cancellation of each excitation contributing to the scattering integral with its particle-hole exchanged counterpart. From the representation in Eqn. (29), we observe that the dynamics on the $(n, m)$ level is proportional to the following factor

$$\mathcal{J}^{(n,m)}(\mathbf{p}_1, \mathbf{h}_1, \mathbf{p}_2, \mathbf{h}_2) = J_1^n(\mathbf{p}_1, \mathbf{h}_1) J_2^m(\mathbf{p}_2, \mathbf{h}_2) - J_1^n(\mathbf{h}_1, \mathbf{p}_1) J_2^m(\mathbf{h}_2, \mathbf{p}_2). \tag{34}$$

The remaining factors under the integral are strictly positive, thus the state in the first tube is stationary if

$$\mathcal{J}^{(n,m)}(\mathbf{p}_1, \mathbf{h}_1, \mathbf{p}_2, \mathbf{h}_2) = 0, \tag{35}$$

holds for all rapidities $\mathbf{p}_1, \mathbf{h}_1, \mathbf{p}_2, \mathbf{h}_2$ satisfying the momentum-energy conservation laws

$$\sum_{i=1}^{n} \big( k_1(p_{i,1}) - k_1(h_{i,1}) \big) + \sum_{j=1}^{m} \big( k_2(p_{j,2}) - k_2(h_{j,2}) \big) = 0, \tag{36}$$

$$\sum_{i=1}^{n} \big( \omega_1(p_{i,1}) - \omega_1(h_{i,1}) \big) + \sum_{j=1}^{m} \big( \omega_2(p_{j,2}) - \omega_2(h_{j,2}) \big) = 0. \tag{37}$$

For stationarity, Eqn. (35) needs to be fulfilled on all $(n, m)$ levels. The same factors, $\mathcal{J}^{(n,m)}$ are present in the collision integral for the second tube and so stationarity in tube 1 implies stationarity in tube 2.

To understand the relevant structure in Eqn. (35), it is useful to reparametrize the system introducing the pseudo energy $\epsilon(\lambda)$ defined as [36, 49]

$$e^{\epsilon(\lambda)} = \frac{\rho_h(\lambda)}{\rho_p(\lambda)}. \tag{38}$$

We may then rewrite $\mathcal{J}^{(n,m)}(\mathbf{p}_1, \mathbf{h}_1, \mathbf{p}_2, \mathbf{h}_2)$ as

$$\mathcal{J}^{(n,m)}(\mathbf{p}_1, \mathbf{h}_1, \mathbf{p}_2, \mathbf{h}_2) = J_1^n(\mathbf{h}_1, \mathbf{p}_1) J_2^m(\mathbf{h}_2, \mathbf{p}_2) \big( \exp[\epsilon_1(\mathbf{p}_1, \mathbf{h}_1) + \epsilon_2(\mathbf{p}_2, \mathbf{h}_2)] - 1 \big), \tag{39}$$

and the condition in Eqn. (35) translates to

$$\sum_{i=1}^{n} \big( \epsilon_1(p_{i,1}) - \epsilon_1(h_{i,1}) \big) + \sum_{j=1}^{m} \big( \epsilon_2(p_{j,2}) - \epsilon_2(h_{j,2}) \big) = 0. \tag{40}$$

Importantly, the state of the system with two tubes in thermal states [36] with the same inverse temperature $\beta$ is stationary. This is because in such a case, the pseudo energies, here generalized to boosted thermal states, read [36, 42]

$$\epsilon_1(\lambda) = -\beta \mu_1 + \kappa k_1(\lambda) + \beta \omega_1(\lambda), \qquad \epsilon_2(\lambda) = -\beta \mu_2 + \kappa k_2(\lambda) + \beta \omega_2(\lambda), \tag{41}$$

where $\mu_{1,2}$ are the chemical potentials. For $\epsilon_{1,2}$ as in Eqn. (41), Eqn. (40) follows directly from the conservation of energy (37) and momentum (36). For all the numerical examples considered later, we set $\kappa = 0$.

The dynamics restricted to $(1,1)$ processes are special and feature stationary states that are not necessarily thermal. This may happen when we consider two identical tubes (this case was analyzed before in [30]), in the Tonks-Girardeau limit in both tubes ($c_{1,2} = \infty$), or, finally in the small momentum limit of the collision integral. This last scenario provides the setting of our observed prethermalization plateaus is the central result of this work. We discuss that case in Section 4 in detail.

In Appendix B we provide evidence that the three types of athermal stationary states of the $(1,1)$ processes are the only ones. However we cannot assert that these are definitively the only possibilities. Moreover for higher order $(n,m)$ processes the condition for the stationarity would seem to exclude any possibility of athermal states. We conjecture once all processes are taken into account that the only stationary states are then in fact thermal.

### 3.3 Conservation laws

The evolution in Eqn. (15) is controlled by a presence of conservation laws. In particular the particle number in each tube, the total momentum, and the total energy do not change in time. In order to see this, let us consider the change in the particle number with respect to time

$$
\begin{aligned}
\partial_t N_1/L &= \int d\lambda\, \partial_t \rho_{p,i}(\lambda) = \int d\lambda\, Q[\rho_{p,1}, \rho_{p,2}](\lambda) \\
&= \int d\lambda\, Q_0[\rho_{p,1}, \rho_{p,2}](\lambda) + \int d\mu d\lambda\, \partial_\lambda\big(n(\lambda)F(\lambda|\mu)\big)Q_0[\rho_{p,1}, \rho_{p,2}](\mu) = 0.
\end{aligned}
\tag{42}
$$

In the last step, we have used the explicit form of (23) together with ph symmetry of the form factor to observe that the first integral vanishes. The second integral vanishes upon integration by parts with the boundary terms vanishing as for physical states the filling function decays to zero at large rapidities. Of course, similar arguments lead to $\partial_t N_2 = 0$. Note that this conservation law holds at all levels $(n,m)$.

Now, let us consider the total energy

$$
E_{tot} = E_1 + E_2 = L \int d\lambda\, \lambda^2\big(\rho_{p,1}(\lambda) + \rho_{p,2}(\lambda)\big),
\tag{43}
$$

and show that it is conserved in time. We consider its derivative with respect to time

$$
\begin{aligned}
\partial_t E_{tot}/L &= \int d\lambda\, \lambda^2\big(Q[\rho_{p,1}, \rho_{p,2}](\lambda) + Q[\rho_{p,2}, \rho_{p,1}](\lambda)\big) \\
&= \int d\lambda d\mu\, \lambda^2\big(\mathbb{F}_1(\lambda,\mu)Q_0[\rho_{p,1}, \rho_{p,2}](\mu) + \mathbb{F}_2(\lambda,\mu)Q_0[\rho_{p,2}, \rho_{p,1}](\mu)\big).
\end{aligned}
\tag{44}
$$

We may shift the dressing operation from the collision integral $Q$ to $\lambda^2$. Integration over $\lambda$ then gives

$$
\int d\lambda\, \lambda^2 \mathbb{F}_i(\lambda,\mu) = \omega_i(\mu).
\tag{45}
$$

We thus get

$$
\partial_t E_{tot}/L = \int d\mu\big(\omega_1(\mu)Q_0[\rho_{p,1}, \rho_{p,2}](\mu) + \omega_2(\mu)Q_0[\rho_{p,2}, \rho_{p,1}](\mu)\big).
\tag{46}
$$

We observe now that the following relation holds for arbitrary $(n,m)$

$$
\int d\mu\big(\omega_1(\mu)Q_0^{(n,m)}[\rho_{p,1}, \rho_{p,2}](\mu) + \omega_2(\mu)Q_0^{(m,n)}[\rho_{p,2}, \rho_{p,1}](\mu)\big) = 0.
\tag{47}
$$

One can establish this by employing the representation in Eqn. (31) into the equation above. From the integration over $\mu$, we find terms that exactly correspond to $\omega_1(\mathbf{p}_1, \mathbf{h}_1) + \omega_2(\mathbf{p}_2, \mathbf{h}_2)$ which is enforced to be zero by the Dirac delta inside $Q_0^{(n,m)}$. Finally, summing this relation above over $n$ and $m$ gives $\partial_t E_{tot} = 0$. Similar arguments lead to the conclusion that the total momentum

$$P_{tot} = P_1 + P_2\,, \tag{48}$$

is conserved as well.

In the remaining part of this section, we propose the following ansatz for additional conserved quantities:

$$\mathcal{I} = L \int d\lambda \left( \rho_{p,1}(\lambda) f_1(\lambda) + \rho_{p,2}(\lambda) f_2(\lambda) \right), \tag{49}$$

where we allowed $f_i(\lambda)$'s, the single particle eigenvalues of the supposed charges, to be different in both tubes. We ask whether there exist functions $f_1(\lambda)$ and $f_2(\lambda)$ such that $\partial_t \mathcal{I} = 0$. Explicitly, we write

$$\begin{aligned}
\partial_t \mathcal{I}/L &= \int d\lambda\, \partial_t \rho_{p,1}(\lambda) f_1(\lambda) + \int d\lambda\, \partial_t \rho_{p,2}(\lambda) f_2(\lambda) \\
&= \int d\lambda \left( f_1(\lambda) Q[\rho_{p,1}, \rho_{p,2}](\lambda) + f_2(\lambda) Q[\rho_{p,2}, \rho_{p,1}](\lambda) \right) \\
&= \int d\lambda \left( f_{1,Dr}(\lambda) Q_0[\rho_{p,1}, \rho_{p,2}](\lambda) + f_{2,Dr}(\lambda) Q_0[\rho_{p,2}, \rho_{p,1}](\lambda) \right),
\end{aligned} \tag{50}$$

where we have shifted the dressing operation from $Q$ to $f_i$ and used the Dressing operation from Eqn. (11). Similarly to the condition for the stationarity, $\partial_t \mathcal{I} = 0$ either because the expression vanishes as a whole or because the integrand is zero. In the latter case this leads to a condition on functions $f_i(\lambda)$, similar to Eqn. (40) which we will now derive.

Analogously to the case of conserved energy, in order to have $\partial_t \mathcal{I} = 0$, we require that

$$\int d\lambda \left( f_{1,Dr}(\lambda) Q_0^{(m,n)}[\rho_{p,1}, \rho_{p,2}](\lambda) + f_{2,Dr}(\lambda) Q_0^{(n,m)}[\rho_{p,2}, \rho_{p,1}](\lambda) \right) = 0, \tag{51}$$

should hold for all $(n, m)$. To write the result in a compact form we parametrise $Q_0^{(m,n)}$ as

$$\begin{aligned}
Q_0^{(n,m)}(\lambda) = \int d\mathbf{p}_1^n d\mathbf{h}_1^n d\mathbf{p}_2^m d\mathbf{h}_2^m\, G(\mathbf{p}_1, \mathbf{h}_1, \mathbf{p}_2, \mathbf{h}_2) \\
\times\, \delta(k_1 + k_2)\, \delta(\omega_1 + \omega_2) \left( \sum_{i=1}^n \delta(\lambda - p_{i,1}) - \sum_{i=1}^n \delta(\lambda - h_{i,1}) \right),
\end{aligned} \tag{52}$$

where $G(\mathbf{p}_1, \mathbf{h}_1, \mathbf{p}_2, \mathbf{h}_2)$ can be read off from Eqn. (31) but we do not need its explicit form. We only need to know that it is strictly positive. The result is

$$\begin{aligned}
\partial_t \mathcal{I} = \int d\mathbf{p}_1^n d\mathbf{h}_1^n d\mathbf{p}_2^m d\mathbf{h}_2^m\, G(\mathbf{p}_1, \mathbf{h}_1, \mathbf{p}_2, \mathbf{h}_2)\, \delta(k_1 + k_2)\, \delta(\omega_1 + \omega_2) \\
\times \left( \sum_{i=1}^n f_{1,Dr}(p_{i,1}) - f_{1,Dr}(h_{i,1}) + \sum_{j=1}^m f_{2,Dr}(p_{j,2}) - f_{2,Dr}(h_{j,2}) \right) = 0\,.
\end{aligned} \tag{53}$$

From this expression we can read off a *necessary* condition for an existence of a conserved charge characterized locally in the rapidities. This translates to demanding that

$$\sum_{i=1}^n \left( f_{1,Dr}(p_{i,1}) - f_{1,Dr}(h_{i,1}) \right) + \sum_{j=1}^m \left( f_{2,Dr}(p_{j,2}) - f_{2,Dr}(h_{j,2}) \right) = 0, \tag{54}$$

is fulfilled for all $\mathbf{p}_1, \mathbf{h}_1, \mathbf{p}_2, \mathbf{h}_2$ satisfying the energy-momentum conservation laws (37) and (36). We observe a similar structure as in Eqn. (40) for the stationary state.[2] Therefore, by exactly the same arguments, we find no conserved charges of the form (49) apart from total momentum and energy. However, just as in the case of stationary state analysis, the situation will be different for $(1,1)$ processes which we discuss in details in the next two sections.

## 4 (1,1) dynamics in the small momentum limit

We have argued that due to the scaling with the characteristic momentum $k_r$, the processes involving small number of ph excitations are the most relevant. In this section, we study in detail the dynamics generated by $(1,1)$ processes solely. As we will show, in the small momentum limit of collision integral, such restricted dynamics are qualitatively different. In particular, the restricted dynamics feature non-thermal stationary states.

### 4.1 Stationary states

We start by reassessing the condition for the stationary state. In the small momentum limit it is convenient to introduce a center-of-mass coordinates for the rapidities,

$$\lambda_i = \frac{1}{2}(p_i + h_i), \qquad \alpha_i = p_i - h_i, \quad i = 1, 2, \tag{55}$$

with $\alpha_i$ being small for small $k = k_i(p_i) - k_i(h_i)$. We analyze now the condition (40) together with the momentum-energy conservation laws (36) and (37) taking $(n, m) = (1, 1)$. After expanding in $\alpha$, the energy-momentum conservation laws give

$$v_1(\lambda_1) = v_2(\lambda_2), \qquad v(\lambda) \equiv \omega'(\lambda)/k'(\lambda), \tag{56}$$

$$\alpha_2 = -k_1'(\lambda_1)/k_2'(\lambda_2)\alpha_1. \tag{57}$$

On the other hand, expansion of (40) in small $\alpha_i$ together with (56) and (57) yield the following condition for the stationary state:

$$\frac{\epsilon_1'(\lambda_1)}{k_1'(\lambda_1)} = \frac{\epsilon_2'(\lambda_2)}{k_2'(\lambda_2)}, \qquad v_1(\lambda_1) = v_2(\lambda_2). \tag{58}$$

Eqn. (58) sets the relation between pseudo energies in both tubes in the stationary state. The relation between velocities allows us to express $\lambda_2$ in terms of $\lambda_1$.[3] Therefore, the equation involving the pseudo energies has only one variable appearing and for any state $\epsilon_2(\lambda)$ it can be used to determine $\epsilon_1(\lambda)$ (up to a constant value). One may readily check that a state in which both tubes are in equally boosted thermal states with the same temperature, see eq. (41), fulfills (58) and therefore is a stationary state in terms of the small momentum limit of the $(1,1)$ dynamics. Note however, that in principle there might be other configurations, for which (58) is satisfied. In the remaining part of the paper, we will show that this is the case and the $(1,1)$ dynamics in the small momentum limit typically drive the system to such stationary but non-thermal states. This feature is particular to $(1,1)$ dynamics. The small momentum expansion of processes involving more ph excitations does not lead to the appearance of non-thermal stationary states.

---

[2]There is however a crucial difference between the two equations - the equation for the conserved charges has to be fulfilled dynamically at each time $t$. We will see the consequences of this in the simplest case of $(1,1)$ processes in Section 5.

[3]We are assuming here that the effective velocity is a monotonic function in $\lambda$.

It is difficult to write down a solution to Eqn. (58) for arbitrary interaction parameters as this requires, for a given $\epsilon_1(\lambda)$, solving a non-linear equation for $\epsilon_2(\lambda_2)$. The non-linearity appears because the functions, $k_2(\lambda_2)$ and $v_2(\lambda_2)$, depend themselves on $\epsilon_2'(\lambda_2)$ through the occupation function, $n_2(\lambda_2)$. Therefore, instead of constructing the solution, we will verify that the system, when evolved according to $(1,1)$ dynamics in the small momentum limit, is described by the pseudo-energies $\epsilon_i(\lambda_i)$ obeying (58). We will also explicitly see that the state is non-thermal. These results are presented in Section 4.2. But to gain some insight, we will now study Eqn. (58) in two special cases where the analysis simplifies:

**Tonks-Girardeau limit in both tubes**

The Tonks-Girardeau limit is particularly simple as the dressings are then absent. From Eqn. (58) we then find

$$\epsilon_1'(\lambda) = \epsilon_2'(\lambda), \tag{59}$$

where we used that $k'(\lambda) = 1$ and $v(\lambda) = 2\lambda$ in the Tonks-Girardeau gas. This implies that pseudo-energies of both tubes in the stationary states are equal up to the chemical potentials: $\epsilon_2(\lambda) = \epsilon_1(\lambda) + \text{const}$. Such solution obviously allows for a much wider class of configurations than (41).

Interestingly, this solution is valid also beyond the small momentum limit. This is a peculiarity of the Tonks-Girardeau gas where the conservations laws (36) and (37) at arbitrary momenta can be solved explicitly and yield $p_2 = h_1$ and $h_2 = p_1$. The condition (40) for the stationary state simplifies then to

$$\epsilon_1(p) - \epsilon_2(p) = \epsilon_1(h) - \epsilon_2(h), \tag{60}$$

and has the same solution as in the small momentum limit.

**Deformed Tonks-Girardeau gas**

Assuming the presence in the two tubes large couplings $c_1$ and $c_2$, we may analyze (58) by expanding in $1/c_i$. Details here are shifted to Appendix D. Summarizing this calculation, we find that the allowed set of pseudo-energies implying stationarity extends beyond that corresponding to boosted thermal states. Indeed, a stationary state is given by any pair of functions $(\epsilon_1\lambda), \epsilon_2(\lambda))$ satisfying,

$$\xi_2^{-1}\epsilon_2'(\xi_2\lambda) = \xi_1^{-1}\epsilon_1'(\xi_1\lambda) \times \left(1 + \mathcal{O}\left(1/c_i^3\right)\right), \quad \xi_i = 1 + \frac{2n_i}{c_i}, \tag{61}$$

which is a simple modification of the stationarity condition (59) in the Tonks-Girardeau gas. Here $n_i$ is the density of particles in the $i$-th tube.

## 4.2 Numerical results

We now numerically solve the Boltzmann equation to demonstrate the main findings of our work. We study relaxation under dynamics restricted to $(1,1)$ processes. We neglect the processes involving more ph pairs effectively considering Eqns. (15) with $Q_0 \equiv Q_0^{(1,1)}$. Moreover, we consider potentials narrow in the momentum space such that $Q_0^{(1,1)}$ is very well approximated by its low-momentum limit. With this, we may expect both non-thermal stationary states (58) as the final states of the evolution.

To display the athermal character of the stationary states we will compare them with the predictions of the standard Gibbs ensemble (GE). The bare pseudoenergies are then

$$\epsilon_{0,i}(\lambda) = \beta_{i,0} + \beta_2 \lambda^2, \tag{62}$$

with $\beta_{1,0}, \beta_{2,0}$ and $\beta_2$ fixed by the initial densities in both tubes and their total energy.

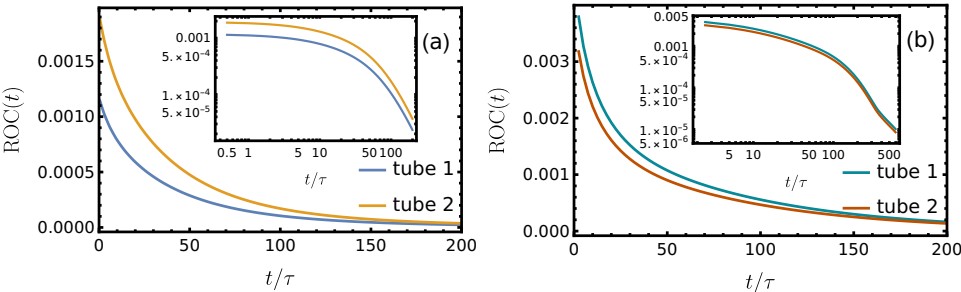

Figure 1: Rate of change of quasi-momentum distributions as a function of time, with Bragg-split states as initial states for the dynamics. Panel ($a$): intermediate regime, panel (b): strongly interacting regime. In both cases the systems reach the stationary state, with a different timescale for each tube. Insets: the same plots, but for longer times and in logarithmic scale.

Table 1: Parameters of the initial states used for the two cases of Bragg-split states explored in this paper.

| case 1 (intermediate interactions) | | | | | | | case 2 (strong interactions) | | | | | | |
|---|---|---|---|---|---|---|---|---|---|---|---|---|---|
| $c_1$ | $c_2$ | $\beta_1$ | $\beta_2$ | $\mu_1$ | $\mu_2$ | $\lambda_0$ | $c_1$ | $c_2$ | $\beta_1$ | $\beta_2$ | $\mu_1$ | $\mu_2$ | $\lambda_0$ |
| 1.0 | 4.0 | 0.7 | 0.7 | 1.6 | 2.6 | 2.0 | 32.0 | 128.0 | 1.7 | 1.7 | 3.22 | 5.27 | 2.9 |

### 4.2.1 Bragg-split states

As initial states, we consider the experimentally relevant Bragg-split thermal distributions [19]. To probe both the intermediate and strong interaction regimes, we perform two computations: one with $c_1 = 1$ and $c_2 = 4$, and one with $c_1 = 32$, $c_2 = 128$. Concretely, we consider initial states of the following form:

$$\rho_{p,i}^{\text{ini}}(\lambda) = \frac{1}{2}\left(\rho_{\beta_i,\mu_i}^{\text{Gibbs}}(\lambda - \lambda_0) + \rho_{\beta_i,\mu_i}^{\text{Gibbs}}(\lambda + \lambda_0)\right), \tag{63}$$

with the parameters used summarized in Table 1. As the first probe of the relaxation, we investigate the rate of change of the quasiparticle distributions as a function of time defined in the following way

$$\text{ROC}(t) = \frac{\int d\lambda |\partial_t \rho_p(\lambda;t)|}{\int d\lambda \rho_p(\lambda;t)}. \tag{64}$$

The results are shown at Fig. 1. We observe that the dynamics becomes gradually slower with time and the system reaches a stationary state. The timescales of these processes are different in each tube. This is expected, since couplings and densities differ between the tubes.

The final, stationary state reached in the course of $(1,1)$ dynamics is presented in Fig. 2. We compare in this figure the final quasiparticle densities $\rho_{p,i}, i = 1, 2$ to the initial distributions as well as to the predictions of the GE ensemble. We observe that the final, stationary state still resembles a Bragg-split distribution and is very far from the thermal GE state that one would expect from unconstrained dynamics. This is further confirmed as one inspects the Yang-Yang entropy as a function of time as shown in Fig. 3. The system relaxes to a state with a much lower entropy in comparison to the prediction computed on the thermal GE state.

Finally, we verify explicitly that the final state fulfills the stationarity condition (58). To do so, we define a function

$$H(\lambda_1) = \rho_{p,1}(\lambda_1)\rho_{h,1}(\lambda_1)\rho_{p,2}(\lambda_2)\rho_{h,1}(\lambda_2)\left(\epsilon_1'(\lambda_1) - \frac{k_1'(\lambda_1)}{k_2'(\lambda_2)}\epsilon_2'(\lambda_2)\right), \tag{65}$$

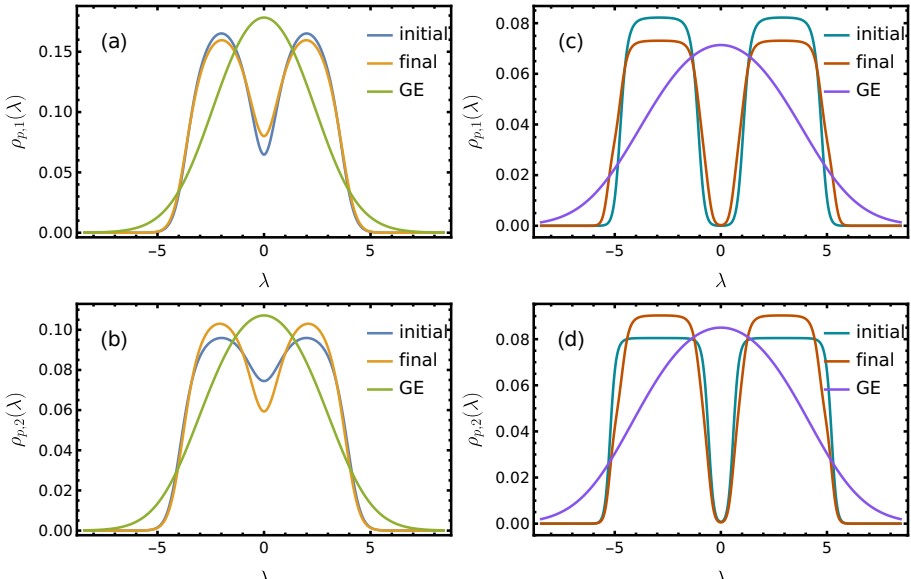

Figure 2: Initial, final and GE states for the dynamics with Bragg-split states as initial states. Panels (a) and (b) present distributions for the simulation in the intermediate regime, whereas in panels (c) and (d) we show the data for the strongly interacting regime. The final states are far from their respective GE states expected to appear as the final, thermal states in a full dynamics involving higher processes.

Table 2: Parameters used for the two cases of initial thermal states.

| case 1 (intermediate interactions) | | | | | | case 2 (strong interactions) | | | | | |
|---|---|---|---|---|---|---|---|---|---|---|---|
| $c_1$ | $c_2$ | $\beta_1$ | $\beta_2$ | $\mu_1$ | $\mu_2$ | $c_1$ | $c_2$ | $\beta_1$ | $\beta_2$ | $\mu_1$ | $\mu_2$ |
| 2.0 | 4.0 | 0.4 | 0.7 | 2.7 | 4.2 | 32.0 | 128.0 | 0.4 | 0.7 | 9.0 | 9.6 |

where $\lambda_2$ is given by a solution to $v_1(\lambda_1) = v_2(\lambda_2)$ and plot it for several instances of time. If our stationary state is indeed a solution to the Eqn. (58), we should see function $H(\lambda_1)$ approaching zero at all $\lambda$ for late times. We find that this is the case, see Fig. 4.

### 4.2.2 Two different thermal states

In the second scenario, to probe dynamics closer to an equilibrium state, we consider initial configurations corresponding to thermal states in both tubes. Specifically, the tubes are taken to have the same density $n_{1,2} = 1$ but different temperatures. The densities of the initial states read

$$\rho_{p,i}^{\text{ini}}(\lambda) = \rho_{\beta_i,\mu_i}^{\text{Gibbs}}(\lambda), \tag{66}$$

with the parameters used summarized in Table 2.

Despite the thermal starting point, the evolution for these states leads to final, non-thermal states that are nonetheless close to thermal states. This is illustrated in Fig. 5 where we compare the distributions $\rho_p(\lambda)$ in the initial and final states and in putative thermal equilibrium states consistent with the initial states. The lack of thermalization can be also witnessed in the evolution of the entropy, see Fig. 6. Here we see that the entropy draws close to the equilibrium thermal entropy but does not quite reach it.

The two kinds of initial states that we have considered show certain universal features of the dynamics generated by the (1,1) processes for long range interactions. We observe that these processes are not very efficient in redistributing the particles in the tubes. On a practical

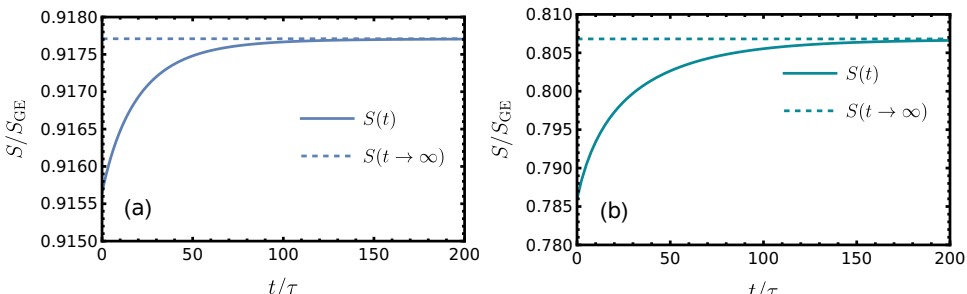

Figure 3: Yang-Yang entropy as a function of time for the dynamics of Bragg-split states. Panel ($a$): simulation in the intermediate regime, panel ($b$) dynamics in the strongly interacting regime. The entropy is normalized by the GE prediction for the final state. It reaches a value significantly smaller than 1, indicating that the dynamics is strongly constrained.

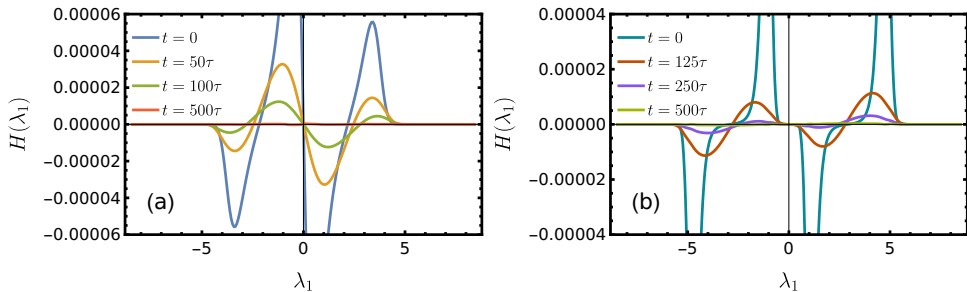

Figure 4: Function $H(\lambda_1)$ presented as a function of time. Panel (a): intermediate interactions, panel ($b$) strong interactions. We see that the function approaches zero for late times signalling that the state of the system in this limit fulfills (58).

level it results in small changes to the particles distributions with the stationary states very similar to the initial states. This is true for both Bragg-split and the thermal initial states. On the other hand, this raises a question on a mechanism behind this restricted dynamics. In the next section we show that in the case of strong intra-tube interactions this is caused by the existence of higher conserved charges.

## 5 Strongly interacting limit

In this section we consider the deformed Tonks-Girardeau gas. This is the strongly interacting limit of large $c_i$ in which we keep corrections up to and including terms of order $1/c_i^2$. We will show that in such system there is an infinite family of conserved charges. These charges follow from solutions of eq. (54) and are linear combination of ultra-local charges (6) present in uncoupled tubes. Whereas the uncoupled system possesses two infinite families of charges, coupling the two tubes leads to halving the number of them. This leads to restricted $(1, 1)$ dynamics observed in the previous section.

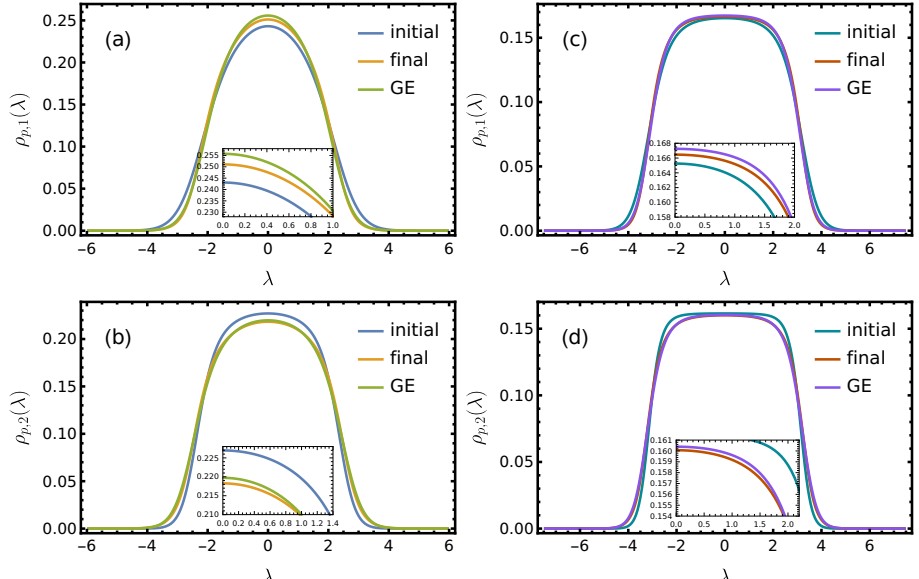

Figure 5: Initial, final and GE states for the dynamics with thermal states as initial states. Panels (a) and (b) present distributions for the simulation in the intermediate regime, whereas in panels (c) and (d) we show the data for strongly interacting regime. Final distributions are very close to the predictions of the GE, indicating that the system has almost thermalized. The small but real differences between the final and putative thermal distributions are clearly seen in the insets, where we have plotted the behavior of the distributions near $\lambda = 0$.

## 5.1 Conserved charges

We start by analyzing the sufficient condition (54) for an existence of a conserved charge. For the $(1,1)$ processes and in the small momentum limit it takes the form

$$\frac{(f_1')_{\mathrm{dr},1}(\lambda_1)}{\omega_1'(\lambda_1)} - \frac{(f_2')_{\mathrm{dr},2}(\lambda_2)}{\omega_2'(\lambda_2)} = 0\,, \qquad v_1(\lambda_1) = v_2(\lambda_2)\,, \tag{67}$$

where we have used the equality in Eqn. (13) between the two dressing procedures. We see that Eqn. (67) has exactly the same structure as Eqn. (58). Therefore, all solutions in the previous section for $\epsilon_{1,2}$ may be adapted for the case considered here. However there is a small caveat here that has far reaching consequences.

To illustrate the nature of the problem let us assume that we choose some function $f_1(\lambda)$ and solve for $f_2(\lambda)$. Because of the dressing procedures it is reasonable to expect that the resulting function depends on the distributions $\rho_{\mathrm{p,i}}(\lambda)$. However, those evolve in time, whereas $f_2(\lambda)$ should be time-independent. Therefore, one possibility is that $f_2(\lambda)$ depends on the particles distributions through invariants of the evolution in each tube and the only such invariants are the total densities $n_i$ of the particles.

This scenario is realized for the deformed Tonks-Girardeau gas. As we show in the Appendix D, Eq. (67) has a time independent solution only up to and including order $1/c_i^2$. The solution takes the form

$$\xi_2^{-1} f_2'(\xi_2 \lambda) = \xi_1^{-1} f_1'(\xi_1 \lambda)\,. \tag{68}$$

Beyond that order it is impossible to find a pair of functions $(f_1(\lambda), f_2(\lambda))$ such that they are time independent and fulfil (67). Note that this solution has exactly the same structure as Eq. (61) for the stationary state of the deformed Tonks-Girardeau gas.

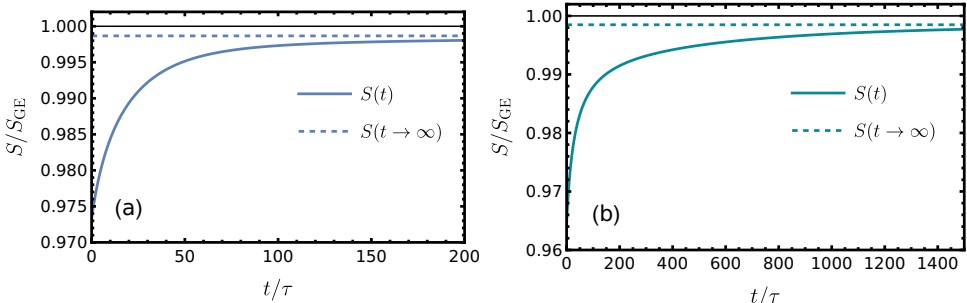

Figure 6: Yang-Yang entropy as a function of time, for the dynamics with initial thermal states. Panel ($a$): simulation in the intermediate regime, panel ($b$) dynamics in the strongly interacting regime. The entropy is normalized by the prediction of the GE ensemble for the final state. The final, stationary state is close to a thermal one, which is reflected in value of normalized entropy that is very close to 1.

From the solution in (68) we can construct a whole family of conserved charges based on the ultra-local conserved charges (6) present in a single Lieb-Liniger model. This results in the following conserved charges

$$\mathcal{I}_j = I_{1,j} + I_{2,j}, \qquad I_{i,j} = L \int d\lambda \, f_{i,j}(\lambda) \rho_{p,i}(\lambda), \qquad f_{i,j}(\lambda) = \xi_i^{2-j} \lambda^j. \tag{69}$$

We see that, due to the $n_i$ dependent coefficients, the resulting expressions are in principle non-linear functions of densities $\rho_{p,i}$. However, since we are working at fixed density they are in fact linear. Even quantum mechanically, in a Hilbert space of fixed number of particles, the corresponding conserved charges would be linear operators. Interestingly, the rescaling of rapidities is the one found also in the $1/c$ expansion of ultra-local conserved charges [38].

The special feature of the dynamics of strongly interacting systems can be directly seen in the collision integral. In Appendix C we show that the collision integral with Tonks-Girardeau gas in both tubes obeys a symmetry relation

$$Q_0^{(1,1)}[\rho_{p,1}, \rho_{p,2}](\lambda) = -Q_0^{(1,1)}[\rho_{p,2}, \rho_{p,1}]((\lambda). \tag{70}$$

This relation immediately implies that any functional of the form (49) with $f_1 = f_2$ is a conserved charge. In the deformed case this relation generalizes to

$$\xi_1^3 Q_0^{(1,1)}[\rho_{p,1}, \rho_{p,2}](\xi_1 \lambda) = -\xi_2^3 Q_0^{(1,1)}[\rho_{p,2}, \rho_{p,1}](\xi_2 \lambda) + \mathcal{O}\left(1/c_i^3\right), \tag{71}$$

and leads to conserved charges of the form (69) as shown in Appendix D.

The integral of motion shown in (69) is conserved only up to $1/c^2$, see Fig. 7. Furthermore, as we show in Appendix E, equation (67) has no solutions that would allow us straightforwardly to construct analytically conserved charges beyond that limit. However, the athermal stationary states do exist for arbitrary values of interaction parameters. This leaves open the question about the mechanism behind the lack of thermalization beyond the strongly interacting limit.

## 5.2 Generalized Gibbs ensemble

In the previous section we have observed that $(1,1)$ dynamics in the small momentum limit and for the deformed Tonks-Girardeau gases is characterized by additional conserved charges $\mathcal{I}_j$. We address now the problem of the final state reached by the system initialized in some

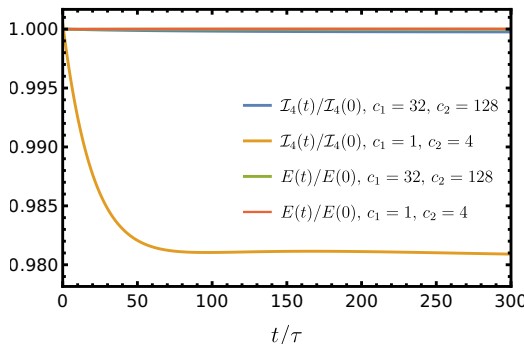

Figure 7: We display the first additional conserved charge $\mathcal{I}_4$ present for the deformed Tonks-Girardeau gas. It is conserved as well as the energies. Evaluating the same functional but in the system with intermediate interactions reveals that it is not anymore a conserved quantity. However the amplitude of its change is small, reflecting again the restricted dynamics of the $(1,1)$ processes. The results shown here are for the Bragg-split states.

non-equilibrium configuration. One would expect relaxation to a Generalized Gibbs Ensemble (GGE) consistent with the presence of non-trivial conservation laws. In this section, we construct a GGE for our system and compare it with the properties of stationary state in Eqn. (58) found from the analysis of the collision integral. The whole construction is a generalization of Ref. [49,50], where a single LL tube was analyzed.

The partition function of our system can be written as a functional integral over the inverse occupation [42],

$$Z = \text{const.} \int \mathcal{D}\left(\frac{\rho_{t,1}(\lambda)}{\rho_{p,1}(\lambda)}\right)\mathcal{D}\left(\frac{\rho_{t,2}(\lambda)}{\rho_{p,2}(\lambda)}\right)\delta(L\rho_{p,1}(\lambda) - N_1)\delta(L\rho_{p,2}(\lambda) - N_2)e^{-\mathcal{F}[\rho_{p,i}]}, \quad (72)$$

where

$$\begin{aligned}
\mathcal{F} &= \beta_2 E + \sum_{j>2}\beta_j \mathcal{I}_j - S \\
&= L\int d\lambda \left[\beta_2 \lambda^2\big(\rho_{p,1}(\lambda) + \rho_{p,2}(\lambda)\big) + \sum_{j>2}\beta_j\big(f_{1,j}(\lambda)\rho_{p,1}(\lambda) + f_{2,j}(\lambda)\rho_{p,2}(\lambda)\big)\right] \\
&\quad - L\sum_{i=1,2}\int d\lambda\left[\rho_{t,i}(\lambda)\ln(\rho_{t,i}(\lambda)) - \rho_{p,i}(\lambda)\ln(\rho_{p,i}(\lambda)) - \rho_{h,i}(\lambda)\ln(\rho_{h,i}(\lambda))\right]. \quad (73)
\end{aligned}$$

In the partition function we have introduced $\delta$-functions that enforce that we are working at constant particle number in each of the tubes. Here we also see the influence that the system's conserved quantities, the energy $E$ and the additional charges, $\mathcal{I}_j, j = 3, 4, \cdots$, preserved under the $(1,1)$ dynamics, have on the partition function. The form of $f_{1/2,j}$ is not arbitrary but assumed to satisfy Eqn. (68). $\beta_2$ is the temperature for the combined system's energy while $\beta_{j>2}$ are the generalized chemical potentials for the higher order charges. Note that the energy and $\mathcal{I}_{j>2}$ are charges defined involving degrees of freedom for both tubes and so $\beta_j$, with $j = 2, \cdots$ do not differentiate between the tubes.

To evaluate the partition function in the thermodynamic limit we use the method of steepest descents [42]. The conserved charges are nonlinear functionals of $\rho_{p,i}$ and their variation with respect to $\rho_{p,i}$ has two contributions

$$\frac{\delta I_{i,j}}{\delta \rho_{p,k}(\lambda)} = \delta_{ik}L\big(f_{i,j}(\lambda) + A_{i,j}\big), \qquad A_{i,j} = \int d\mu \frac{\delta f_{i,j}(\mu)}{\delta \rho_{p,i}(\lambda)}\rho_{p,i}(\mu) = 2(2-j)\frac{I_{i,j}}{L\zeta_i c_i}. \quad (74)$$

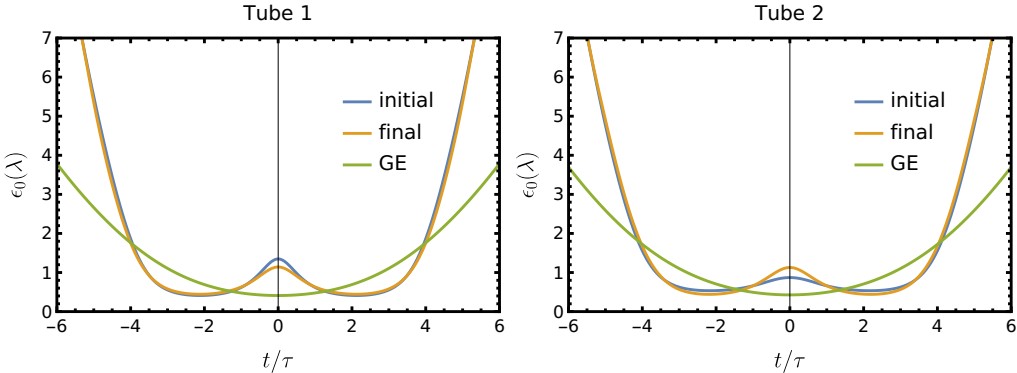

Figure 8: The pseudo-energies for the initial, final and GE states for the Bragg-split dynamics with $c_1 = 1$, $c_2 = 4$.

In writing this expression we have used eq. (69) for $I_{i,j}$ for which $f_{i,j}(\lambda) = \xi_i^{2-j}\lambda^j$.

We introduce now chemical potentials $\beta_{i,0}$ as Lagrange multipliers fixing the total densities of particles in both tubes. Introducing $\epsilon_i(\lambda) = \log(\rho_{h,i}(\lambda)/\rho_{p,i}(\lambda))$, the extremum condition for the generalized free energy is

$$\epsilon_i(\lambda) = \epsilon_{i,0}(\lambda) - \int d\lambda' T_i(\lambda - \lambda') \log\left[1 + e^{-\epsilon(\lambda')}\right], \tag{75}$$

$$\epsilon_{i,0}(\lambda) = \beta_{i,0} + \sum_{j>2}\beta_j A_{i,j} + \beta_2\lambda^2 + \sum_j \beta_j f_{i,j}(\lambda), \tag{76}$$

$$\int d\lambda \rho_{p,i} = n_i, \tag{77}$$

which should be supplemented with the equation for $\rho_{t,i}(\lambda)$,

$$\rho_{t,i}(\lambda) = \frac{1}{2\pi} + \int d\lambda' T(\lambda - \lambda')n_i(\lambda')\rho_{t,i}(\lambda'), \tag{78}$$

which connects $\epsilon_i(\lambda)$ with $\rho_{p,i}(\lambda)$. We observe that while formulas (75) and (76) bear structural similarity to the standard TBA equations, there exists a significant distinction. In a standard formulation, the TBA equations are explicit, meaning that given a set of the chemical potentials, they directly yield $\epsilon(\lambda)$ from which expectation values of the conserved charges can be computed. Here, the relation is implicit, because $\epsilon(\lambda)$ depends itself on the value of the conserved charges through $A_{i,j}$. Nevertheless, the dependence on $A_{i,j}$ enters with a factor $1/c_i$ which allows for a consistent perturbative treatment. For instance, the $1/c_i$ correction to $\epsilon(\lambda)$ comes from the Tonks-Girardeau values of the conserved charges.

The implicit nature of the found GGE is also visible in the expectation values of the charges. The formulas for the particle number and energy are standard and given by: [49]

$$\langle N_i \rangle_{GGE} = \frac{L}{2\pi}\int d\lambda \frac{1}{1 + e^{\epsilon_i(\lambda)}}\frac{\partial \epsilon_i(\lambda)}{\partial \beta_{i,0}}, \qquad \langle E_{tot} \rangle_{GGE} = \frac{L}{2\pi}\sum_{i=1,2}\int d\lambda \frac{1}{1 + e^{\epsilon_i(\lambda)}}\frac{\partial \epsilon_i(\lambda)}{\partial \beta_2}, \tag{79}$$

whereas for the other charges,

$$\langle \mathcal{I}_j \rangle_{GGE} = \sum_{i=1,2}\left[\frac{L}{2\pi}\int d\lambda \frac{1}{1 + e^{\epsilon_i(\lambda)}}\frac{\partial \epsilon_i(\lambda)}{\partial \beta_j} + A_{i,j}\langle N_i \rangle_{GGE}\right]. \tag{80}$$

The values of generalized chemical potentials in the final state are in principle determined from the equations $\langle \cdot \rangle_{ini} = \langle \cdot \rangle_{GGE}$ written for all conserved charges, where by $\langle \cdot \rangle_{ini}$ we denoted

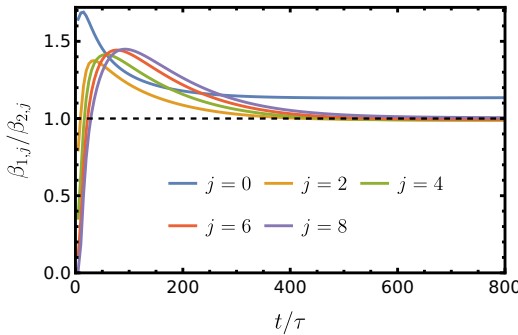

Figure 9: Time evolution of the generalized chemical potentials for the Bragg initial state with $c_1 = 32, c_2 = 128$. We plot the ratio $\beta_{1,j}/\beta_{2,j}$ of the potentials in the two tubes. The system equilibrates to $\beta_{1,j} = \beta_{2,j}$ $j \geq 2$ in agreement with the prediction of the GGE.

the expectation value in the initial state [13]. Unfortunately, solving these non-linear equations is a very difficult task. However, we can show that GGE states of the form (76) are stationary states in our dynamics. The proof of this statement is as follows. We note that

$$\epsilon'_i = (\epsilon'_{0,i})_{\mathrm{dr,i}} \, . \tag{81}$$

Dressing is a linear operation and therefore we can express $\epsilon'_i$ through the dressed single-particle eigenvalues, namely

$$\epsilon'_i = \beta_2 (2\lambda)_{dr,i} + \sum_j \beta_j (f'_{i,j})_{dr,i} \, . \tag{82}$$

In the stationary state the pseudo-energies obey relation (58)

$$\frac{\epsilon'_1(\lambda_1)}{k'_1(\lambda_1)} - \frac{\epsilon'_2(\lambda_2)}{k'_2(\lambda_2)} = \sum_j \beta_j \left( \frac{(f'_{1,j})_{dr,1}(\lambda_1)}{k'_1(\lambda_1)} - \frac{(f'_{2,j})_{dr,2}(\lambda_2)}{k'_2(\lambda_2)} \right) = 0 \, , \tag{83}$$

where in the last step we have used the conservation of the $j$-th conserved charged expressed in Eqn. (67). The athermal character of the pseudo-energies of the stationary states is shown in Fig. 8.

Finally, let us make the following observation. For a single tube Lieb-Liniger model there is a one-to-one relation between the state of the system and chemical potentials (generalized temperatures). We can now promote this relation to the two-tube case. The state of the system is then described by pseudo-energies $\epsilon_i(\lambda)$ characterized by the generalized temperatures $\beta_{i,j}$ which are potentially different for the two tubes, $\beta_{1,j} \neq \beta_{2,j}$. When the system evolves according to the Boltzmann equation this leads to a time dependence of the chemical potentials. The stationary state can be then understood as equalization of temperatures and higher chemical potentials, namely $\lim_{t\to\infty} \left( \beta_{1,j}(t) - \beta_{2,j}(t) \right) = 0$ for $j \geq 2$. We demonstrate it in Fig. 9.

## 6 Summary and conclusion

In this work we have studied the problem of the development of stationary states in a weakly perturbed integrable model. Our approach relies on a microscopic scattering integral in the framework of a Boltzmann equation. We applied this approach to the case of two weakly coupled Lieb-Liniger models, with the coupling taking the form of the density-density interactions,

a scenario relevant for the experiments with ultra-cold atomic gases. We found that for the long-range intertube interactions the leading processes described by the scattering integral do not thermalize the system, instead they lead to an athermal stationary state. To further understand the resulting time evolution we have solved numerically the Boltzmann equation for two very different initial states: i) the highly non-equilibrium Bragg split states and i) tubes characterized by thermal states at different temperatures. In both cases we have observed that time evolution is greatly restricted. The final stationary states resemble the initial states and do not evolve into thermal states. We also witnessed this arrested dynamics through studying the entropy production. There we saw that the system's entropy would evolve to a value lower than that of a thermal ensemble.

For the deformed Tonks-Girardeau gas we have shown that the restricted dynamics is caused by extra conserved charges beyond the particle number, momentum and energy. We have also shown that the athermal stationary state can be then characterized by the generalized Gibbs Ensemble. Interestingly, our construction of the conserved charges does not extend to the full Lieb-Liniger model.

In this work we have focused on revealing the structure of the athermal stationary states. An interesting question would be to combine the leading $(1,1)$ with higher processes that do thermalize the system. This would provide an access to the full time evolution from the initial state, through the prethermalization plateau, all the way to the final thermal state. Numerical implementation of this problem requires a substantial effort that we delegate to future work.

Our work leads also to a more fundamental question whether the extra conserved charges exist at the quantum level, beyond the Boltzmann equation. Whereas their exact conservation is unlikely, they might correspond to 'slow variables' and therefore they remain important for the dynamics at intermediate timescales. This scenario is realized in the spin ladder systems as revealed recently [51].

Finally, the results for the case where the two tubes are initialized in thermal states at different temperatures shows that the evolution occurs entirely in the vicinity of thermal states. In such a situation it might be possible to develop an effective description of the dynamics which instead of looking at the full distribution of the rapidities would involve only examining the evolution of the temperatures and perhaps several of the higher chemical potentials. This would then allow the development of a sort of Newton's law of cooling for weakly perturbed integrable systems.

# Acknowledgments

MP acknowledges insightful discussions with Giuseppe Mussardo, Jacek Herbrych, Marcin Mierzejewski and Jakub Pawłowski.

**Funding information** M.Ł and M.P. were supported by the National Science Centre, Poland, under the SONATA grant 2018/31/D/ST3/03588. R.M.K. was supported by the U.S. Department of Energy, Office of Basic Energy Sciences, under Contract No. DE-SC0012704.

# A  Scaling of the collision integral in the small momentum limit

In this appendix we show that $Q_0^{(m,n)} \sim k_r^{m+n}$ with $k_r$ the characteristic momentum of the intertube potential. From the symmetry of Eqn. (29), we can replace sum of delta functions

by a single one, multiplied by $n$

$$
Q_0^{(n,m)}[\rho_{p,1},\rho_{p,2}](\lambda) = (2\pi)^2 n \int d\mathbf{p}_1^n d\mathbf{h}_1^n d\mathbf{p}_2^m d\mathbf{h}_2^m \, \delta(\lambda - p_{1,1}) \tilde{A}^2\big(k_1(\mathbf{p}_1,\mathbf{h}_1)\big)
$$
$$
\times |F_1(\mathbf{p}_1,\mathbf{h}_1)|^2 |F_2(\mathbf{p}_2,\mathbf{h}_2)|^2 \mathcal{J}^{(n,m)}(\mathbf{p}_1,\mathbf{h}_1,\mathbf{p}_2,\mathbf{h}_2)
$$
$$
\times \delta\big(k_1(\mathbf{p}_1,\mathbf{h}_1) + k_2(\mathbf{p}_2,\mathbf{h}_2)\big) \delta\big(\omega_1(\mathbf{p}_1,\mathbf{h}_1) + \omega_2(\mathbf{p}_2,\mathbf{h}_2)\big),
$$

where we recall the definition of $\mathcal{J}^{(n,m)}$

$$
\mathcal{J}^{(n,m)}(\mathbf{p}_1,\mathbf{h}_1,\mathbf{p}_2,\mathbf{h}_2) = J_1^n(\mathbf{p}_1,\mathbf{h}_1) J_2^m(\mathbf{p}_2,\mathbf{h}_2) - J_1^n(\mathbf{h}_1,\mathbf{p}_1) J_2^m(\mathbf{h}_2,\mathbf{p}_2). \tag{A.1}
$$

We express the collision integrals in the center of mass variables, namely,

$$
\lambda_{i,j} = \frac{1}{2}(p_{i,j} + h_{i,j}), \qquad \alpha_{i,j} = p_{i,j} - h_{i,j}. \tag{A.2}
$$

We make now the following observations. The form-factors of the density operator, due to their ph symmetry are even function of $\alpha_{i,j}$ and their leading order is $\alpha_{i,j}$ independent, i.e.,

$$
F_1(\mathbf{p}_1,\mathbf{h}_1) = F_1(\boldsymbol{\lambda}_1,\boldsymbol{\lambda}_1) + \mathcal{O}(\alpha_i^2). \tag{A.3}
$$

The difference of the density factors appearing through the $J$ functions, see (24), is an odd function upon changing the sign of all $\alpha_{i,j}$. At the same time, the product of the density functions evaluated for a particle-hole excitation gives

$$
\rho_p(\lambda - \alpha/2)\rho_h(\lambda + \alpha/2)\big(1 + \epsilon'(\lambda)\alpha/2 + \mathcal{O}(\alpha^2)\big). \tag{A.4}
$$

Therefore we get

$$
\mathcal{J}^{(n,m)}(\mathbf{p}_1,\mathbf{h}_1,\mathbf{p}_2,\mathbf{h}_2) = \prod_{i=1}^{n} \rho_{p,1}(\lambda_{i,1})\rho_{h,1}(\lambda_{i,1}) \prod_{j=1}^{m} \rho_{p,2}(\lambda_{j,2})\rho_{h,2}(\lambda_{j,2})
$$
$$
\times \left( \sum_{i=1}^{n} \epsilon_1'(\lambda_{1,i})\alpha_{1,i} + \sum_{i=1}^{m} \epsilon_2'(\lambda_{2,i})\alpha_{2,i} + \mathcal{O}(\alpha_{i,j}^3) \right).
$$

The scattering integral, up to the first two leading orders in $\alpha_{i,j}$, takes then the following form,

$$
Q_0^{(n,m)}[\rho_{p,1},\rho_{p,2}](\mu) = (2\pi)^2 n \int d\boldsymbol{\lambda}_1^n d\boldsymbol{\lambda}_2^m G(\boldsymbol{\lambda}_1,\boldsymbol{\lambda}_2) H(\boldsymbol{\lambda}_1,\boldsymbol{\lambda}_2), \tag{A.5}
$$

where

$$
G(\boldsymbol{\lambda}_1,\boldsymbol{\lambda}_2) = |F_1(\boldsymbol{\lambda}_1,\boldsymbol{\lambda}_1)|^2 |F_2(\boldsymbol{\lambda}_2,\boldsymbol{\lambda}_2)|^2 \prod_{i=1}^{n} \rho_{p,1}(\lambda_{i,1})\rho_{h,1}(\lambda_{i,1}) \prod_{j=1}^{m} \rho_{p,2}(\lambda_{j,2})\rho_{h,2}(\lambda_{j,2}), \tag{A.6}
$$

$$
H(\boldsymbol{\lambda}_1,\boldsymbol{\lambda}_2) = \int d\boldsymbol{\alpha}_1^n d\boldsymbol{\alpha}_2^m \tilde{A}^2(k_1(\mathbf{p}_1,\mathbf{h}_1)) \delta(\mu - \lambda_{1,1} - \alpha_{1,1}/2) \delta\big(k_1(\mathbf{p}_1,\mathbf{h}_1) + k_2(\mathbf{p}_2,\mathbf{h}_2)\big)
$$
$$
\times \delta\big(\omega_1(\mathbf{p}_1,\mathbf{h}_1) + \omega_2(\mathbf{p}_2,\mathbf{h}_2)\big) \left( \sum_{i=1}^{n} \epsilon_1'(\lambda_{1,i})\alpha_{1,i} + \sum_{i=1}^{m} \epsilon_2'(\lambda_{2,i})\alpha_{2,i} \right). \tag{A.7}
$$

Consider first the simplest case of $(1,1)$ processes. Then we have to perform the following integral

$$
H(\lambda_1,\lambda_2) = \int d\alpha_1 d\alpha_2 \tilde{A}^2(k_1(\lambda_1,\alpha_1)) \delta(\mu - \lambda_1 - \alpha_1/2) \delta\big(k_1(\lambda_1,\alpha_1) + k_2(\lambda_2,\alpha_2)\big)
$$
$$
\times \delta\big(\omega_1(\lambda_1,\alpha_1) + \omega_2(\lambda_2,\alpha_2)\big) \big(\epsilon_1'(\lambda_1)\alpha_1 + \epsilon_2'(\lambda_2)\alpha_2\big). \tag{A.8}
$$

For the momentum energy constraints we find

$$k(\lambda, \alpha) = \alpha k'(\lambda) + \mathcal{O}(\alpha^3), \qquad \omega(\lambda, \alpha) = \alpha \omega'(\lambda) + \mathcal{O}(\alpha^3). \tag{A.9}$$

Thus we have

$$\delta\big(k_1(\lambda_1, \alpha_1) + k_2(\lambda_2, \alpha_2)\big) = \delta(\alpha_1 k_1'(\lambda_1) + \alpha_2 k_2'(\lambda_2)), \tag{A.10}$$

and similarly for the energy $\delta$-function. Then we can write

$$\delta\big(k_1(\lambda_1, \alpha_1) + k_2(\lambda_2, \alpha_2)\big)\delta\big(\omega_1(\lambda_1, \alpha_1) + \omega_2(\lambda_2, \alpha_2\big)$$
$$= \frac{\delta(\alpha_2 + \alpha_1 k_1'(\lambda_1)/k_2'(\lambda_2))}{|\alpha_1 k_1'(\lambda_1) k_2'(\lambda_2)|}\delta(v_1(\lambda_1) - v_2(\lambda_2)), \tag{A.11}$$

where $v(\lambda) = \omega'(\lambda)/k'(\lambda)$. We can perform now the integral over $\alpha_2$. The result is

$$H(\lambda_1, \lambda_2) = \int \mathrm{d}\alpha_1 \, \mathrm{sgn}(\alpha_1)\tilde{A}^2(k_1(\lambda_1, \alpha_1))\delta(\mu - \lambda_1 - \alpha_1/2)$$
$$\times \frac{\delta(v_1(\lambda_1) - v_2(\lambda_2))}{|k_2'(\lambda_2)|}\left(\frac{\epsilon_1'(\lambda_1)}{k_1'(\lambda_1)} - \frac{\epsilon_2'(\lambda_2)}{k_2'(\lambda_2)}\right). \tag{A.12}$$

The remaining integral over $\alpha_1$ can be now also performed. The result is

$$H(\lambda_1, \lambda_2) = \mathrm{sgn}(\mu - \lambda_1)\tilde{A}^2(k_1(\lambda_1, 2(\mu - \lambda_1)))\frac{\delta(v_1(\lambda_1) - v_2(\lambda_2))}{|k_2'(\lambda_2)|}\left(\frac{\epsilon_1'(\lambda_1)}{k_1'(\lambda_1)} - \frac{\epsilon_2'(\lambda_2)}{k_2'(\lambda_2)}\right). \tag{A.13}$$

The remaining $\delta$-function fixes uniquely $\lambda_2$ as a function of $\lambda_1$. Therefore the scattering integral involves only one integral. The range of the integration over $\lambda_1$ is constrained by the function $\tilde{A}(k_1(\lambda_1, 2(\mu - \lambda_1)))$ which is peaked around $\lambda_1 = \mu$. The structure of the expression is then

$$Q_0^{(1,1)}[\rho_{p,1}, \rho_{p,2}](\mu) = \int \mathrm{d}\lambda_1 \mathrm{sgn}(\mu - \lambda_1)\tilde{A}^2(2(\mu - \lambda_1)k'(\lambda_1))f(\lambda_1), \tag{A.14}$$

where in $f(\lambda_1)$ we have gathered all the remaining factors. Now we will take advantage of properties of the interaction potential and use representation (21). We get

$$Q_0^{(1,1)}[\rho_{p,1}, \rho_{p,2}](\mu) \sim \int \mathrm{d}\lambda_1 \, \mathrm{sgn}(\mu - \lambda_1)\tilde{a}^2\left(\frac{2(\mu - \lambda_1)k'(\lambda_1)}{k_r}\right)f(\lambda_1). \tag{A.15}$$

We assume now that function $\tilde{a}(k/k_r)$ is centered around 0 and symmetric. In the leading order in $k_r$ we can approximate this expression by replacing $k'(\lambda_1)$ by $k'(\mu)$. We then change the variables to $x = 2(\mu - \lambda_1)k'(\mu)/k_r$. The result is

$$Q_0^{(1,1)}[\rho_{p,1}, \rho_{p,2}](\mu) \sim \frac{k_r}{2k'(\mu)} \int \mathrm{d}x \, \mathrm{sgn}(x)\tilde{a}^2(x)f\left(\mu - x\frac{k_r}{2k'(\mu)}\right). \tag{A.16}$$

Expanding now in powers of $x$ we find

$$Q_0^{(1,1)}[\rho_{p,1}, \rho_{p,2}](\mu) \sim \frac{k_r f(\mu)}{2k'(\mu)} \int \mathrm{d}x \, \mathrm{sgn}(x)\tilde{a}^2(x) + \frac{k_r^2 f'(\mu)}{(2k'(\mu))^2} \int \mathrm{d}x \, |x|\tilde{a}(x) + \mathcal{O}(k_r^4). \tag{A.17}$$

The first integral vanishes and the leading contribution comes from the second term which is proportional to the characteristic momentum $k_r$. The answer involves the derivative of

$f(\mu)$. This function involves all the density factors and form-factors. Its derivative is thus quite complicated and not very practical for numerical computations. We also note that the subleading corrections are of order $k_r^4$ and not $k_r^3$ which is important for the hierarchy of processes contributing to the Boltzmann equation as discussed in the main text.

We generalize now the analysis to the $(n,m)$ case. We start by considering $H(\boldsymbol{\lambda}_1, \boldsymbol{\lambda}_2)$ and use the energy-momentum constraints to solve for $\alpha_{m,2}$ and $\lambda_{m,2}$ and the third $\delta$-function to fix $\lambda_{1,1} = \mu - \alpha_{1,1}/2$. To this end we introduce the following notation

$$\bar{k} = \sum_{i=1}^{n} \alpha_{i,1} k_1'(\lambda_{i,1}) + \sum_{i=1}^{m-1} \alpha_{i,2} k_2'(\lambda_{i,2}),\tag{A.18}$$

$$\bar{\omega} = \sum_{i=1}^{n} \alpha_{i,1} \omega_1'(\lambda_{i,1}) + \sum_{i=1}^{m-1} \alpha_{i,2} \omega_2'(\lambda_{i,2}),\tag{A.19}$$

$$\bar{\epsilon} = \sum_{i=1}^{n} \alpha_{i,1} \epsilon_1'(\lambda_{i,1}) + \sum_{i=1}^{m-1} \alpha_{i,2} \epsilon_2'(\lambda_{i,2}),\tag{A.20}$$

and $\bar{v} = \bar{\omega}/\bar{k}$. Then

$$\delta\big(k_1(\mathbf{p}_1, \mathbf{h}_1) + k_2(\mathbf{p}_2, \mathbf{h}_2)\big)\delta\big(\omega_1(\mathbf{p}_1, \mathbf{h}_1) + \omega_2(\mathbf{p}_2, \mathbf{h}_2)\big) = \frac{\delta(\alpha_{m,2} - \bar{\alpha}_{m,2})\delta(v_2(\lambda_{m,2}) - \bar{v})}{k'(\lambda_{m,2})|\bar{k}|},\tag{A.21}$$

with $\bar{\alpha}_{m,2} = \bar{k}/k_2'(\lambda_{m,2})$. This gives

$$H(\boldsymbol{\lambda}_1, \boldsymbol{\lambda}_2) = \int d\boldsymbol{\alpha}_1^n d\boldsymbol{\alpha}_2^{m-1} \tilde{A}^2(k_1(\mathbf{p}_1, \mathbf{h}_1)) \frac{\delta(v_2(\lambda_{1,2}) - \bar{v})}{k'(\lambda_{1,2})} \left(\frac{\bar{\epsilon}}{\bar{k}} - \frac{\epsilon_2'(\lambda_{1,2})}{k_2'(\lambda_{1,2})}\right) \text{sgn}(\bar{k}).\tag{A.22}$$

We now introduce new variables $x_{i,j} = \alpha_{i,j} k'(\lambda_{i,j})/k_r$ for all $i, j$ except for $i = j = 1$ where we write instead $x_{1,1} = \alpha_{1,1} k'(\mu))/k_r$. We now find for $H(\boldsymbol{\lambda}_1, \boldsymbol{\lambda}_2)$,

$$H(\boldsymbol{\lambda}_1, \boldsymbol{\lambda}_2) = k_r^{n+m-1} \frac{\delta(v_2(\lambda_{1,2}) - \bar{v})}{\prod_{i,j} k'(\lambda_{i,j})} \int d\mathbf{x}_1^n d\mathbf{x}_2^{m-1} \text{sgn}(\sum_{i,j} x_{i,j}) \tilde{a}^2\left(\sum_i x_{i,1}\right)\left(\frac{\bar{\epsilon}}{\bar{k}} - \frac{\epsilon_2'(\lambda_{m,2})}{k_2'(\lambda_{m,2})}\right).\tag{A.23}$$

In writing the argument of $\tilde{a}$ we have neglected contributions of order $x_{1,1}^2$ as they do not contribute to the integral in the first two leading orders. The ratio $\bar{\epsilon}/\bar{k}$ is the following

$$\frac{\bar{\epsilon}}{\bar{k}} = \frac{\sum x_{i,j} \epsilon_j'(\lambda_{i,j})/k_j'(\lambda_{i,j}) + x_{1,1} \epsilon_1'(\mu - \alpha_{1,1}/2)/k_1'(\mu))}{\sum x_{i,j} + x_{1,1} k_1'(\mu - \alpha_{1,1}/2)/k_1'(\mu)},\tag{A.24}$$

with $\alpha_{1,1} = k_r x_{1,1}/k_1'(\mu)$. Thus, at the leading order in $k_r$ it is $k_r$ independent and invariant under simultaneous change of sign of all the $x_{i,j}$. As a result, the integral over all the $x_{i,j}$'s vanishes. The leading contribution comes then from including the correction due to $\alpha_{1,1}$. Therefore, the whole contribution to the scattering integral is of the order $k_r^{n+m}$ as stated in the main text.

## B  Absence of non-thermal stationary states beyond the small momentum limit

In the main body of the text we demonstrated that if the dynamics of the two tubes of gas are controlled by $(1,1)$ processes in the small momentum limit, a prethermalization plateau

is arrived at that is characterized by a set of non-trivial conservation laws. In this appendix, we argue that this is in general a special case, that if we consider $(1,1)$ processes outside a small momentum approximation, thermalization occurs. We note up front that there are two exceptions to this general rule:

- The Tonks-Girardeau (TG) limit. The thermalization that occurs due to the higher momentum $(1,1)$ processes happens because of non-linearities in the dispersion relation induced by interactions. But at $c = \infty$ these interactions are absent and the argument for a non-trivial prethermalization plateau continues to hold even if higher momentum processes are accounted for. Moreover we can conclude that TG gases do not thermalize at all as higher-order ph processes are absent. Though we have broken the integrability at the level of a single tube, new exact integrals of motion have appeared for the two-tube case.

- The two gases are identical. In this case the $(1,1)$ processes induces no dynamics whatsoever and the gas does not evolve. Thermalization however will still happen on account of higher order $(n,m)$ processes.

We will comment on these two special cases further.

So let us turn to $(1,1)$ processes at arbitrary momentum. The condition for the stationary state, $Q_0^{(1,1)} = 0$, translates to demanding that

$$\epsilon_1(p_1) - \epsilon_1(h_1) + \epsilon_2(p_2) - \epsilon_2(h_2) = 0, \tag{B.1}$$

for all $p_1, h_1, p_2, h_2$ fulfilling the momentum-energy constraints

$$\begin{aligned}
k_1(p_1) - k_1(h_1) + k_2(p_2) - k_2(h_2) &= 0, \\
\omega_1(p_1) - \omega_1(h_1) + \omega_2(p_2) - \omega_2(h_2) &= 0.
\end{aligned} \tag{B.2}$$

Here functions $k_i(\lambda)$ and $\omega_i(\lambda)$ are determined from the $\epsilon_i(\lambda)$ through the TBA equations. We start by analyzing the structure of Eqns. (B.1) and (B.2).

The constraints (B.2) can be solved for $p_1$ and $h_1$. We assume that the solution is unique and takes the form $p_1 = f(p_2, h_2)$ and $h_1 = g(p_2, h_2)$ for two functions $f$ and $g$. Because of the symmetry of the constraints upon replacing particles with holes the two functions are not independent and we have $g(p_2, h_2) = f(h_2, p_2)$. Therefore, a generic solution to the constraints can be written as

$$p_1 = f(p_2, h_2), \quad h_1 = f(h_2, p_2). \tag{B.3}$$

We define now a function $F(p_2, h_2)$,

$$F(p_2, h_2) \equiv \epsilon_1(f(p_2, h_2)) - \epsilon_1(f(h_2, p_2)). \tag{B.4}$$

The condition (B.1) for the stationary state in these terms is simply

$$F(p_2, h_2) = \epsilon_2(p_2) - \epsilon_2(h_2). \tag{B.5}$$

To obtain a general solution here we assume that $f(p_2, h_2)$ is independent of one of its variables, which we choose to be $p_2$. This simplifies the structure of $F$ to

$$F(p_2, h_2) = \epsilon_1(f(h_2)) - \epsilon_1(f(p_2)). \tag{B.6}$$

The equation for the stationary states separates now in two parts depending only on $p_2$ and $h_2$ respectively. This leads to two independent equations of the same form which fixes the

relation between $\epsilon_1$ and $\epsilon_2$ up to an additive constant which is just a difference of chemical potentials. The result is

$$\epsilon_1(f(\lambda)) = \epsilon_2(\lambda) + \mu_1 - \mu_2. \tag{B.7}$$

The function $f(\lambda)$ by construction satisfies

$$k_1(f(p_2)) - k_1(f(h_2)) + k_2(p_2) - k_2(h_2) = 0, \tag{B.8}$$
$$\omega_1(f(p_2)) - \omega_1(f(h_2)) + \omega_2(p_2) - \omega_2(h_2) = 0. \tag{B.9}$$

Assuming the monotonicity of the dressed momenta $k_i$, it is straightforward to solve the first equation for $f$. The result is $f = k_1^{-1} \circ k_2$. The second equation provides now a constraint between the functions $k_i$ and $\omega_i$ themselves. It reads

$$\omega_2 \circ k_2^{-1} = \omega_1 \circ k_1^{-1}. \tag{B.10}$$

Putting it all together we find the following set of equations

$$\epsilon_1 \circ k_1^{-1}(p) - \mu_1 = \epsilon_2(p) \circ k_2^{-1}(p) - \mu_2,$$
$$\omega_1 \circ k_1^{-1}(p) = \omega_2 \circ k_2^{-1}(p). \tag{B.11}$$

This is a sought after rewriting of equations (B.1) and (B.2). For a given function $\epsilon_2$, the TBA equations determine $k_2$ and $\omega_2$. Thus the right hand sides of the above two equations are known. The equations then have to be solved for $\epsilon_1$ such that TBA integral equations defining $k_1$ and $\omega_1$ in terms of $\epsilon_1$ are self-consistent.

We show now three special cases where we can solve the above:

- *Thermal states:* For a thermal state in the second tube, $\epsilon_2(\lambda) = \mu_2 + \beta \omega_2(\lambda)$, we have a chain of following transformations

$$\epsilon_1 \circ k_1^{-1}(p) - \mu_1 = \epsilon_2 \circ k_2^{-1}(p) - \mu_2 \tag{B.12}$$
$$= \beta \omega_2 \circ k_2^{-1}(p) \tag{B.13}$$
$$= \beta \omega_1 \circ k_1^{-1}(p). \tag{B.14}$$

  Therefore the solution is $\epsilon_1(\lambda) = \mu_1 + \beta \omega_1(\lambda)$ and we know that this choice is consistent with the TBA integrals defining $\omega_1$ and $k_1$.

- *Identical gases in identical states:* If the two tubes of gas are identical in all respects, then any choice of $\epsilon_1 = \epsilon_2$ (both thermal and athermal) satisfies the above conditions. Because $k_1 = k_2$ and $\omega_1 = \omega_2$, any scattering process must have $p_1 = h_2$ and $p_2 = h_1$. By symmetry under exchange of particles and holes, $Q^{1,1}$ vanishes identically and at this order, the gas does not evolve from its initial state.

- *Tonks-Girardeau gas:* When both tubes are in the strongly interacting limit, $c_i = \infty$, then there is no dressing and the second equality in (B.11) is trivially fulfilled. At the same time the first equations simplifies to $\epsilon_1(\lambda) - \mu_1 = \epsilon_2(\lambda) - \mu_2$. Therefore the stationary state is given by the pseudo-energies equal up to the chemical potentials which can be different in both tubes.

We cannot prove that other solutions are impossible, but we were unable to find additional possibilities. Therefore we expect that in general the higher momentum contributions to the $(1,1)$ processes lead to the thermalization of the system.

Further support of this contention is provided by considering an expansion of the stationary state condition in the momentum transferred between the states. To this end we expand eqs. (B.1) and (B.2) in $\alpha_i = p_i - h_i$. At leading order we obtain the following constraint:

$$
\begin{aligned}
v_1(\lambda_1) &= v_2(\lambda_2), \\
\epsilon_1'(\lambda_1) &= \frac{k_1'(\lambda_1)}{k_2'(\lambda_2)}\epsilon_2'(\lambda_2),
\end{aligned}
\tag{B.15}
$$

where $\lambda_i = \frac{1}{2}(p_i + h_i)$. We already know that these constraints admit solutions that are athermal. This was our main result in the main body of the text. At next order in $\alpha_i$, $\mathcal{O}(\alpha_i^3)$, we obtain additional constraints that need to be satisfied for stationarity:

$$
\begin{aligned}
v_2(\lambda_1) &= \left(1 - \frac{k_1'(\lambda_1)^3}{k_2'(\lambda_2)^3}\frac{k_2'''(\lambda_2)}{k_1'''(\lambda_1)}\right)^{-1}\left(\frac{\omega_1'''(\lambda_1)}{k_1'''(\lambda_1)} - \frac{k_1'(\lambda_1)^3}{k_2'(\lambda_2)^3}\frac{\omega_2'''(\lambda_2)}{k_1'''(\lambda_1)}\right), \\
\epsilon_1'''(\lambda_1) &= \left(\frac{k_1'''(\lambda_1)}{k_2'(\lambda_2)} - \frac{k_1'(\lambda_1)^3}{k_2'(\lambda_2)^4}k_2'''(\lambda_2)\right)\epsilon_2'(\lambda_2) + \frac{k_1'(\lambda_1)^3}{k_2'(\lambda_2)^3}\epsilon_2'''(\lambda_2).
\end{aligned}
\tag{B.16}
$$

It seems unlikely that both constraints (B.15) and (B.16) can be satisfied simultaneously, beyond the specialized circumstances discussed above. A similar line of reasoning may be straightforwardly extended to processes involving more ph pairs.

## C Tonks-Girardeau gas

In this Appendix we study in details the Boltzmann equation for the Tonks-Girardeau gas, $c \to \infty$. In this limit the (1,1) processes are the only processes because the form-factors with larger number of particle-hole pairs vanish. Also in this limit the dressings are absent what significantly reduces the complexity of the problem and allows us to write the scattering integral in a simple form.

We start with (29) adopted to (1,1) processes. Using that in the Tonks-Girardeau (TG) gas the form-factors of single particle-hole excitations are identically 1 we obtain

$$
\begin{aligned}
Q^{(1,1)}[12](\lambda) = (2\pi)^2 \int dp_1 dh_1 dp_2 dh_2\, \delta(\lambda - p_1)\tilde{A}^2\big(k_1(p_1, h_1)\big) \\
\times \delta\big(k_1(p_1, h_1) + k_2(p_2, h_2)\big)\delta\big(\omega_1(p_1, h_1) + \omega_2(p_2, h_2)\big) \\
\times (J_1(p_1, h_1)J_2(p_2, h_2) - J_1(h_1, p_1)J_2(h_2, p_2)),
\end{aligned}
\tag{C.1}
$$

In this and next Appendix we shorten the notation from $Q^{(1,1)}[\rho_{p,1}, \rho_{p,2}]$ to $Q^{(1,1)}[12]$. The momentum and energy in TG gas are given by $k(\lambda) = \lambda$ and $\omega(\lambda) = \lambda^2$ respectively. This allows us to solve the kinematic constraint $k(p_1, h_1) + k(p_2, h_2) = 0$ and $\omega(p_1, h_1) + \omega(p_2, h_2) = 0$. In terms of the center-of-mass rapidities $\lambda_i = (p_i + h_i)/2$ and $\alpha_i = p_i - h_i$ we find

$$
\delta(k(p_1, h_1) + k(p_2, h_2))\delta(\omega(p_1, h_1) + \omega(p_2, h_2)) = \frac{\delta(\alpha_1 + \alpha_2)\delta(\lambda_1 - \lambda_2)}{2\alpha_i}.
\tag{C.2}
$$

The Jacobian of transformation from $(p_i, h_i)$ to $(\lambda_i, \alpha_i)$ is 1 and the collision integral, after evaluating integrals over $\lambda_2$ and $\alpha_2$ with the help of the Dirac $\delta$-functions, becomes

$$
Q_0^{(1,1)}[12](\lambda) = (2\pi)^2 \int d\lambda_1 d\alpha_1 \delta(\lambda - p_1)\frac{\tilde{A}^2(\alpha_1)}{2|\alpha_1|}(J_1(p_1, h_1)J_2(p_2, h_2) - J_1(h_1, p_1)J_2(h_2, p_2)).
\tag{C.3}
$$

Evaluating now the integral over $\lambda_1$ and dropping the index of $\alpha_1$ gives

$$Q^{(1,1)}[12](\lambda) = (2\pi)^2 \int d\alpha \left(J_1(\lambda, \lambda - \alpha)J_2(\lambda - \alpha, \lambda) - J_1(\lambda + \alpha, \lambda)J_2(\lambda, \lambda + \alpha)\right) \frac{\tilde{A}^2(\alpha)}{2|\alpha|}.$$

(C.4)

This is the final expression for the collision integral in the TG gas. It shows that the dynamics is driven by the difference in particles distributions with the rate controlled by the coupling potential.

The collision integral $Q^{(1,1)}[21](\lambda)$ for the second tube comes from exchanging the indices 1 and 2 and $Q^{(1,1)}[21](\lambda) = -Q^{(1,1)}[12](\lambda)$. This implies that the following combination

$$\int d\lambda \left(\rho_{p,1}(\lambda) + \rho_{p,2}(\lambda)\right) f(\lambda),$$

(C.5)

is a conserved charge for any $f(\lambda)$ such that the integral exists.

We can also find the stationary state. From Eqn. (C.3) the condition is

$$J_1(\lambda, \lambda - \alpha)J_2(\lambda - \alpha, \lambda) - J_1(\lambda - \alpha, \lambda)J_2(\lambda, \lambda - \alpha) = 0,$$

(C.6)

which gives $\epsilon_1(\lambda) = \epsilon_2(\lambda) + \text{const}$ in agreement with Eqn. (59) from the main text.

In the limit of small momentum transfer between the tubes, the collision integral can be further evaluated by expanding the integrand in $\alpha$. The density factors have the following small $\alpha$ expansion

$$J_i(\lambda + \alpha, \lambda) = \rho_{p,i}(\lambda)\rho_{h,i}(\lambda)\left(1 + \frac{\rho'_{h,i}(\lambda)}{\rho_{h,i}(\lambda)}\alpha + \frac{1}{2}\frac{\rho''_{h,i}(\lambda)}{\rho_{h,i}(\lambda)}\alpha^2\right),$$

(C.7)

$$J_i(\lambda, \lambda + \alpha) = \rho_{p,i}(\lambda)\rho_{h,i}(\lambda)\left(1 + \frac{\rho'_{p,i}(\lambda)}{\rho_{p,i}(\lambda)}\alpha + \frac{1}{2}\frac{\rho''_{p,i}(\lambda)}{\rho_{p,i}(\lambda)}\alpha^2\right).$$

(C.8)

After substituting these expansions, we find that the leading term vanishes identically under the integral while the term linear in $\alpha$ vanishes after integrating over $\alpha$ ($\tilde{A}(\alpha)$ is an even function). Therefore, the leading contribution comes from order $\alpha^2$ and is given by

$$Q_0^{(1,1)}[12](\lambda) = (2\pi)^2 \left(\prod_{i=1,2} \rho_{p,i}(\lambda)\rho_{h,i}(\lambda)\right) G_{1,2}(\lambda) \int d\alpha \frac{|\alpha|\tilde{A}^2(\alpha)}{4},$$

(C.9)

with

$$G_{1,2}(\lambda) = \frac{\rho''_{p,1}(\lambda)}{\rho_{p,1}(\lambda)} - \frac{\rho''_{h,1}(\lambda)}{\rho_{h,1}(\lambda)} - \frac{\rho''_{p,2}(\lambda)}{\rho_{p,2}(\lambda)} + \frac{\rho''_{h,2}(\lambda)}{\rho_{h,2}(\lambda)} + 2\left(\frac{\rho'_{p,1}(\lambda)\rho'_{h,2}(\lambda)}{\rho_{p,1}(\lambda)\rho_{h,2}(\lambda)} - \frac{\rho'_{h,1}(\lambda)\rho'_{p,2}(\lambda)}{\rho_{h,1}(\lambda)\rho_{p,2}(\lambda)}\right).$$

(C.10)

We can now extract the $k_r$ dependence of the collision integral. We write $\tilde{A}(\alpha) = \tilde{a}(\alpha/k_r)$ according to Eqn. (21) and change the integration variable to find that $Q_0^{(1,1)}[12](\lambda) \sim k_r^2$ in agreement with the general result presented in Appendix A.

# D  Deformed Tonks-Girardeau gas

In this Appendix we consider the deformed Tonks-Girardeau, the $c \to \infty$ limit in which we keep terms $1/c$ up to and including order $1/c^2$. In practice, we will keep also terms of higher orders. This will be useful for Appendix E to illustrate a qualitative difference between the deformed Tonks-Girardeau gas and systems with weaker interactions.

## TBA for the deformed Tonks-Girardeau gas

We start with the relation between the filling function $n(\theta)$ and the particles' density $\rho_{\mathrm{p}}(\lambda)$. First, we use the defining relation (2) between $\rho_{\mathrm{p}}(\lambda)$ and $\rho_{\mathrm{t}}(\lambda)$,

$$\rho_t(\lambda) = \frac{1}{2\pi} + \int \mathrm{d}\mu\, T(\lambda - \mu)\rho_p(\mu). \tag{D.1}$$

The kernel $T(\lambda - \mu)$, defined in Eqn. (3), in the large $c$ limit expands to

$$T(\lambda - \mu) = \frac{1}{\pi c}\left(1 - \left(\frac{\lambda - \mu}{c}\right)^2 + \left(\frac{\lambda - \mu}{c}\right)^4\right) + \mathcal{O}\left(1/c^7\right). \tag{D.2}$$

Substituting this expansion in Eqn. (D.1) and integrating term by term gives

$$\rho_t(\lambda) = \frac{1}{2\pi}\left(1 + \frac{2n}{c} - \frac{2}{c^3}\left(\lambda^2 n + e\right) + \frac{2}{c^5}\left(\lambda^4 n + 6\lambda^2 e + q_4\right) + \mathcal{O}\left(1/c^7\right)\right), \tag{D.3}$$

where we defined

$$q_i = \int \mathrm{d}\mu\, \mu^i \rho_{\mathrm{p}}(\mu), \tag{D.4}$$

with $n = q_0$ and $e = q_2$. The filling function then follows

$$n(\lambda) = \frac{\rho_p(\lambda)}{\rho_t(\lambda)} = 2\pi\rho_p(\lambda)\left(1 - \frac{2n}{c} + \frac{4n^2}{c^2} + \frac{2(n\lambda^2 + e - 4n^3)}{c^3} + \mathcal{O}\left(1/c^4\right)\right). \tag{D.5}$$

Consider now the back-flow function. It obeys the following equation (we reproduce here Eqn. (10))

$$F(\lambda|\mu) = \frac{\theta(\lambda - \mu)}{2\pi} + \int \mathrm{d}\lambda'\, T(\lambda - \lambda')n(\lambda')F(\lambda'|\mu), \tag{D.6}$$

and in the large $c$ expansion can be solved iteratively. This procedure yields

$$F(\lambda|\mu) = \frac{\lambda - \mu}{\pi c} - \frac{(\lambda - \mu)^3}{3\pi c^3} - \frac{\mu}{(\pi c)^2}\int \mathrm{d}\lambda'\, n(\lambda')\left(1 + \frac{1}{\pi c}\int \mathrm{d}\lambda'\, n(\lambda')\right) + \mathcal{O}\left(1/c^4\right). \tag{D.7}$$

The integrals over the filling function can be evaluated. We will need only the first two leading corrections

$$\int \mathrm{d}\lambda\, n(\lambda) = 2\pi n\left(1 - \frac{2n}{c} + \frac{4n^2}{c^2}\right) + \mathcal{O}\left(1/c^3\right). \tag{D.8}$$

With this expression the back-flow function simplifies to

$$F(\lambda|\mu) = \frac{\lambda - \mu}{\pi c} - \frac{2n\mu}{\pi c^2} - \frac{(\lambda - \mu)^3}{3\pi c^3} + \mathcal{O}\left(1/c^4\right). \tag{D.9}$$

With the knowledge of the back-flow function we can now compute the Dressed momentum and energy. We recall the formulas (9a) and (9b) from which we obtain

$$k(\lambda) = \lambda\left(1 + \frac{2n}{c}\right) - \frac{2\lambda}{c^3}\left(\frac{\lambda^2 n}{3} + e\right) + \mathcal{O}\left(1/c^4\right), \tag{D.10}$$

$$\omega(\lambda) = \lambda^2\left(1 + \frac{4e}{c^3}\right) + \mathrm{const} + \mathcal{O}\left(1/c^4\right). \tag{D.11}$$

The final ingredient of the TBA that we need is the dressing operation. This is defined in Eqn. (12). Expanding again the kernel produces

$$f_{\mathrm{dr}}(\lambda) = f(\lambda) + \frac{1}{\pi c} \int d\mu \left( 1 - \left( \frac{\lambda - \mu}{c} \right)^2 + \left( \frac{\lambda - \mu}{c} \right)^4 + \dots \right) n(\mu) f_{\mathrm{dr}}(\mu), \qquad \text{(D.12)}$$

which is solved by

$$f_{\mathrm{dr}}(\lambda) = f(\lambda) + \frac{1}{\pi c} \frac{\int d\mu\, n(\mu) f(\mu)}{1 - \frac{1}{\pi c} \int d\mu\, n(\mu)} - \frac{1}{\pi c^3} \int d\mu\, (\lambda - \mu)^2 n(\mu) f(\mu) + \mathcal{O}\left( 1/c^5 \right). \quad \text{(D.13)}$$

Expressing now the filling function $n(\lambda)$ by $\rho_{\mathrm{p}}(\lambda)$ using Eqn (D.5), we find the final formula

$$f_{\mathrm{dr}}(\lambda) = f(\lambda) + \frac{2}{c} \int d\mu \rho_p(\mu) f(\mu) - \frac{2}{c^3} \int d\mu (\lambda - \mu)^2 \rho_p(\mu) f(\mu) + \mathcal{O}\left( 1/c^4 \right). \qquad \text{(D.14)}$$

This result implies the following expansions for $k' = (1)_{\mathrm{dr}}$ and $\omega' = (2\lambda)_{\mathrm{dr}}$,

$$k'(\lambda) = 1 + \frac{2n}{c} - \frac{2}{c^3} \left( \lambda^2 n + e \right) + \mathcal{O}\left( 1/c^4 \right), \qquad \text{(D.15)}$$

$$\omega'(\lambda) = 2\lambda \left( 1 + \frac{4e}{c^3} + \mathcal{O}\left( 1/c^4 \right) \right), \qquad \text{(D.16)}$$

which are in agreement with (D.10) and (D.11). The effective velocity then follows

$$v(\lambda) = \frac{\omega'(\lambda)}{k'(\lambda)} = 2\lambda \left( 1 - \frac{2n}{c} + \frac{4n^2}{c^2} + \frac{2n\lambda^2 + 6e - 8n^3}{c^3} + \mathcal{O}\left( 1/c^4 \right) \right). \qquad \text{(D.17)}$$

Here, we summarize the formulas for the deformed Tonks-Girardeau gas which will be important in the computation of the collision integral. For the deformed Tonks-Girardeau gas (neglecting contributions of order $1/c^3$ and higher) we find

$$k(\lambda) = \xi k_{\mathrm{TG}}(\lambda), \qquad \text{(D.18)}$$

$$\omega(\lambda) = \omega_{\mathrm{TG}}(\lambda) + \mathrm{const}, \qquad \text{(D.19)}$$

where we defined

$$\xi = 1 + \frac{2n}{c}. \qquad \text{(D.20)}$$

We also note that the back-flow function of the deformed Tonks-Girardeau gas has a simple form

$$F(\lambda|\mu) = \frac{\lambda - \xi\mu}{\pi c} + \mathcal{O}\left( 1/c^3 \right). \qquad \text{(D.21)}$$

**Collision integral**

We compute now the collision integral for $(1,1)$ processes in the deformed Tonks-Girardeau gas. This means including the corrections of order $1/c_i^2$. However, the computations simplify and their structure is more apparent if we include also contributions of higher orders. Specifically, we will often write $\xi_i^{-1}$ and not expand it. This results in expressions that formally contain contributions of higher orders but are correct only up to and including orders $1/c_i^2$.

We define the rescaled center-of-mass coordinates

$$p_i = \xi_i \lambda_i + \alpha_i/(2\xi_i), \qquad h_i = \xi_i \lambda_i - \alpha_i/(2\xi_i). \qquad \text{(D.22)}$$

The Jacobian of transformation from $(p_i, h_i)$ to $(\lambda_i, \alpha_i)$ is 1. We express now the kinematic constraint appearing in the collision integral in terms of these new coordinates. To this end, we use (D.18) and (D.19) to find

$$\delta\big(k_1(p_1, h_1) + k_2(p_2, h_2)\big)\delta\big(\omega_1(p_1, h_1) + \omega_2(p_2, h_2)\big) = \frac{\delta(\alpha_1 + \alpha_2)\delta(\lambda_1 - \lambda_2)}{2|\alpha_1|}. \tag{D.23}$$

We will use this expression in the formula for $Q_0^{(1,1)}[12](\lambda)$ which for the convenience we repeat here

$$Q^{(1,1)}[12](\lambda) = (2\pi)^2 \int dp_1 dh_1 dp_2 dh_2\, \delta(\lambda - p_1)\tilde{A}^2\big(k_1(p_1, h_1)\big)|F(p_1, h_1)|^2 |F(p_2, h_2)|^2$$
$$\times \delta\big(k_1(p_1, h_1) + k_2(p_2, h_2)\big)\delta\big(\omega_1(p_1, h_1) + \omega_2(p_2, h_2)\big)$$
$$\times (J_1(p_1, h_1)J_2(p_2, h_2) - J_1(h_1, p_1)J_2(h_2, p_2)). \tag{D.24}$$

Changing the integration variables from $(p_i, h_i)$ to $(\lambda_i, \alpha_i)$ and performing the integrals over $\lambda_2$ and $\alpha_2$, with the help of the $\delta$-functions from the kinematic constraints, sets $\lambda_2 = \lambda_1$ and $\alpha_2 = -\alpha_1$. The result is

$$Q_0^{(1,1)}[12](\lambda) = (2\pi)^2 \int d\lambda_1 d\alpha_1 \delta(\lambda - p_1)\frac{\tilde{A}^2(\alpha_1)}{2|\alpha_1|}|F(p_1, h_1)|^2 |F(p_2, h_2)|^2$$
$$\times (J_1(p_1, h_1)J_2(p_2, h_2) - J_1(h_1, p_1)J_2(h_2, p_2)) + \mathcal{O}\big(1/c_i^3\big), \quad \text{(D.25)}$$

with

$$\begin{aligned} p_1 &= \xi_1\lambda_1 + \alpha_1/(2\xi_1), & h_1 &= \xi_1\lambda_1 - \alpha_1/(2\xi_1), \\ p_2 &= \xi_2\lambda_1 - \alpha_1/(2\xi_2), & h_2 &= \xi_2\lambda_1 + \alpha_1/(2\xi_2). \end{aligned} \tag{D.26}$$

The analysis simplifies if we evaluate the collision integral at $\xi_1\lambda$ instead of $\lambda$. We resolve then the remaining Dirac $\delta$-function with the result

$$Q_0^{(1,1)}[12](\xi_1\lambda) = (2\pi)^2 \xi_1^{-1} \int d\alpha \frac{\tilde{A}^2(\alpha)}{2|\alpha|}|F(p_1, h_1)|^2 |F(p_2, h_2)|^2$$
$$\times (J_1(p_1, h_1)J_2(p_2, h_2) - J_1(h_1, p_1)J_2(h_2, p_2)) + \mathcal{O}\big(1/c_i^3\big), \tag{D.27}$$

where

$$\begin{aligned} p_1 &= \xi_1\lambda, & h_1 &= \xi_1\lambda - \alpha/\xi_1, \\ p_2 &= \xi_2\lambda - \left(\frac{\xi_1}{\xi_2} + \frac{\xi_2}{\xi_1}\right)\frac{\alpha}{2\xi_1}, & h_2 &= \xi_2\lambda + \left(\frac{\xi_1}{\xi_2} - \frac{\xi_2}{\xi_1}\right)\frac{\alpha}{2\xi_1}. \end{aligned} \tag{D.28}$$

This is the final formula for the collision integral in the deformed Tonks-Girardeau gas valid at any momentum transfer between the tubes.

We can compute now the approximated collision integral valid at small momenta transfer between the tubes by expanding the integrand in small $\alpha$. The computations are similar to the ones in the Tonks-Girardeau gas. The extra ingredient is the form factor which we first approximate in the small momentum limit by $F(p, h) = k'(\lambda) + \mathcal{O}(\alpha^2)$ and then use $k'(\lambda) = \xi + \mathcal{O}\big(1/c^3\big)$. Thus, the only effect of the form-factor is in rescaling of $Q_0^{(1,1)}[12](\lambda)$ by factor $\xi_1^2\xi_2^2$. The final answer is

$$Q_0^{(1,1)}[12](\xi_1\lambda) = (2\pi)^2 \xi_1^{-1}\xi_2^2\left(\prod_{i=1,2}\rho_{p,i}(\xi_i\lambda)\rho_{h,i}(\xi_i\lambda)\right)G_{1,2}(\xi_1\lambda)\int d\alpha \frac{|\alpha|\tilde{A}^2(\alpha)}{4} + \mathcal{O}\big(1/c_i^3\big), \tag{D.29}$$

with

$$
\begin{aligned}
G_{1,2}(\xi_1\lambda) = {} & \frac{\rho''_{p,1}(\xi_1\lambda)}{\rho_{p,1}(\xi_1\lambda)} - \frac{\rho''_{h,1}(\xi_1\lambda)}{\rho_{h,1}(\xi_1\lambda)} - \frac{\rho''_{p,2}(\xi_2\lambda)}{\rho_{p,2}(\xi_2\lambda)} + \frac{\rho''_{h,2}(\xi_2\lambda)}{\rho_{h,2}(\xi_2\lambda)} \\
& + \left(\frac{\xi_1}{\xi_2} + \frac{\xi_2}{\xi_1}\right)\left(\frac{\rho'_{p,1}(\xi_1\lambda)\rho'_{h,2}(\xi_2\lambda)}{\rho_{p,1}(\xi_1\lambda)\rho_{h,2}(\xi_2\lambda)} - \frac{\rho'_{h,1}(\xi_1\lambda)\rho'_{p,2}(\xi_2\lambda)}{\rho_{h,1}(\xi_1\lambda)\rho_{p,2}(\xi_2\lambda)}\right) \\
& + \left(\frac{\xi_1}{\xi_2} - \frac{\xi_2}{\xi_1}\right)\left(\frac{\rho'_{p,1}(\xi_1\lambda)\rho'_{p,2}(\xi_2\lambda)}{\rho_{p,1}(\xi_1\lambda)\rho_{p,2}(\xi_2\lambda)} - \frac{\rho'_{h,1}(\xi_1\lambda)\rho'_{h,2}(\xi_2\lambda)}{\rho_{h,1}(\xi_1\lambda)\rho_{h,2}(\xi_2\lambda)}\right).
\end{aligned} \tag{D.30}
$$

We observe the symmetry $G_{1,2}(\xi_1\lambda) = -G_{2,1}(\xi_2\lambda)$, which implies that

$$
\xi_1^3 Q_0^{(1,1)}[12](\xi_1\lambda) = -\xi_2^3 Q_0^{(1,1)}[21](\xi_2\lambda) + \mathcal{O}\left(1/c_i^3\right). \tag{D.31}
$$

In the deformed Tonks-Girardeau gas to compute the collision integral we need to include the effect of the Dressing. The Dressing of the collision integral is given by the action of the operator, see Eqn. (17),

$$
\mathbb{F}_i(\lambda,\mu) = \delta(\lambda-\mu) + \partial_\lambda(n_i(\lambda)F_i(\lambda|\mu)). \tag{D.32}
$$

In the deformed Tonks-Girardeau gas it becomes

$$
\mathbb{F}_i(\lambda,\mu) = \delta(\lambda-\mu) + \frac{1}{\pi c}\partial_\lambda(\lambda n_i(\lambda)) - \frac{2\mu}{c}\partial_\lambda\rho_p(\lambda) + \mathcal{O}\left(1/c^3\right). \tag{D.33}
$$

The dressed scattering integral is then

$$
Q^{(1,1)}[12](\lambda) = (\mathbb{F}\cdot Q_0)(\lambda) = Q_0^{(1,1)}[12](\lambda) + Q_1^{(1,1)}[12](\lambda) + \mathcal{O}\left(1/c_i^3\right), \tag{D.34}
$$

with

$$
Q_1^{(1,1)}[12](\lambda) = \frac{1}{\pi c}\partial_\lambda(\lambda n_i(\lambda))\int \mathrm{d}\mu\, Q_0^{(1,1)}[12](\mu) - \partial_\lambda\rho_{p,1}(\lambda)\frac{2}{c_1}\int \mathrm{d}\mu\, \mu Q_0^{(1,1)}[12](\mu). \tag{D.35}
$$

The first integral is 0 which reflects the conservation of the total particle number, see Eqn. (42).

## Conserved charges

Consider now a candidate for an integral of motion,

$$
\mathcal{I} = \int \mathrm{d}\lambda\left(f_1(\lambda)\rho_{p,1}(\lambda) + f_2(\lambda)\rho_{p,2}(\lambda)\right), \tag{D.36}
$$

with $f_i(\lambda)$ an even function, otherwise the integral is identically zero. The time evolution of $\mathcal{I}$ has two contributions. The contribution from the bare collision integral $Q_0$ is

$$
\dot{\mathcal{I}}_{\mathrm{direct}} = \int \mathrm{d}\lambda\left(f_1(\lambda)Q_0[12](\lambda) + f_2(\lambda)Q_0[21](\lambda)\right). \tag{D.37}
$$

We change the integration variable to $\lambda = \xi_1\bar{\lambda}$ in the contribution from the first tube and to $\lambda = \xi_2\bar{\lambda}$ in the contribution from the second tube. We obtain

$$
\dot{\mathcal{I}}_{\mathrm{direct}} = \int \mathrm{d}\bar{\lambda}\left(\xi_1 f_1(\xi_1\bar{\lambda})Q_0[12](\xi_1\bar{\lambda}) + \xi_2 f_2(\xi_2\bar{\lambda})Q_0[21](\xi_2\bar{\lambda})\right). \tag{D.38}
$$

Using the symmetry (D.31) of the collision integral we find the following condition for $\dot{J}_0 = 0$,

$$\xi_1^{-2} f_1(\xi_1 \bar{\lambda}) = \xi_2^{-2} f_2(\xi_2 \bar{\lambda}). \tag{D.39}$$

This implies that for arbitrary $n \geq 2$

$$f_i(\lambda) = \xi_i^{-n+2} \lambda^n, \tag{D.40}$$

gives a quantity that does not evolve under the bare collision integral.

The second contribution to $\dot{Q}$ comes from the indirect collision integral $Q_1[12]$,

$$\dot{\mathcal{I}}_{\text{indirect}} = \int d\lambda \left( f_1'(\lambda) \rho_{p,1}(\lambda) \frac{2}{c_1} \int d\mu\, \mu Q_0[12](\mu) + f_2'(\lambda) \rho_{p,2}(\lambda) \frac{2}{c_2} \int d\mu\, \mu Q_0[21](\mu) \right). \tag{D.41}$$

This contribution vanishes because $\rho_{p,i}(\lambda)$ are even function whereas $f_i'(\lambda)$ is odd.

**Stationary state**

We consider now the equation for the stationary state. It takes the universal form, valid for any value of the interaction parameters,

$$\frac{\epsilon_1'(\lambda_1)}{\omega_1'(\lambda_1)} - \frac{\epsilon_2'(\lambda_2)}{\omega_2'(\lambda_2)} = 0, \qquad v_1(\lambda_1) = v_2(\lambda_2). \tag{D.42}$$

We start by solving $v_1(\lambda_1) = v_2(\lambda_2)$.

We use the large $c$ expansion of the effective velocity given in (D.17). This results in the following equation

$$\xi_1^{-1} \lambda_1 = \xi_2^{-1} \lambda_2, \tag{D.43}$$

We can now solve for $\epsilon_2'(\lambda_2)$. The result is

$$\epsilon_2'(\lambda_2) = \epsilon_1'(\xi_1 \xi_2^{-1} \lambda_2) \frac{\omega_2'(\lambda_2)}{\omega_1'(\xi_1 \xi_2^{-1} \lambda_2)} = \xi_1^{-1} \xi_2 \epsilon_1'(\xi_1 \xi_2^{-1} \lambda_2), \tag{D.44}$$

where in the second step we used that $\omega'(\lambda) = 2\lambda$ in the deformed Tonks-Girardeau gas. The resulting equation can be presented in a more symmetric way as

$$\xi_2^{-1} \epsilon_2'(\xi_2 \lambda) = \xi_1^{-1} \epsilon_1'(\xi_1 \lambda). \tag{D.45}$$

# E Stationary states and the possibility of conserved charges beyond the deformed Tonks-Girardeau gas

In this Appendix we construct solutions to the stationary state equation in the $1/c$ expansion that go beyond the deformed Tonks-Girardeau limit by including corrections of order $\mathcal{O}\left(1/c^3\right)$. We, similarly, attempt the construction of the conserved charges to order $\mathcal{O}\left(1/c^3\right)$. Here, however, we show that it is not possible in our framework to find solutions to this order.

**Stationary state**

We start with the stationary state equation. We use the large $c$ expansion of the effective velocity given in (D.17). This results in the following equation

$$\xi_1^{-1} \lambda_1 (1 + 3A_1 + B_1 \lambda_1^2) = \xi_2^{-1} \lambda_2 \left(1 + 3A_2 + B_2 \lambda_2^2\right), \tag{E.1}$$

where

$$A_i = \frac{2e_i}{c_i^3}, \qquad B_i = \frac{2n_i}{c_i^3}. \tag{E.2}$$

Its perturbative solution is

$$\lambda_1 = \frac{\xi_1}{\xi_2}\lambda_2\left(1 + 3(A_2 - A_1) + \lambda_2^2(B_2 - B_1)\right) + \mathcal{O}\left(1/c_i^4\right). \tag{E.3}$$

We can now solve for $\epsilon_2'(\lambda_2)$. The result is

$$\epsilon_2'(\lambda_2) = \epsilon_1'(\lambda_1(\lambda_2))\frac{\omega_2'(\lambda_2)}{\omega_1'(\lambda_1(\lambda_2))}. \tag{E.4}$$

The ratio of the energies, using Eqn. (D.16), becomes

$$\frac{\omega_2'(\lambda_2)}{\omega_1'(\lambda_1(\lambda_2))} = \frac{\xi_2}{\xi_1}\left(1 + A_1 - A_2 + (B_1 - B_2)\lambda_2^2\right). \tag{E.5}$$

We find

$$\epsilon_2'(\lambda_2) = \frac{\xi_2}{\xi_1}\epsilon_1'\left(\frac{\xi_1}{\xi_2}\lambda_2\right) + 2\left(\frac{e_1}{c_1^3} - \frac{e_2}{c_2^3} + \left(\frac{n_1}{c_1^3} - \frac{n_2}{c_2^3}\right)\lambda_2^2\right)\left(\epsilon_1'(\lambda_2) - \lambda_2\epsilon_1''(\lambda_2)\right)$$
$$- 4\left(\frac{e_1}{c_1^3} - \frac{e_2}{c_2^3}\right)\lambda_2\epsilon_1''(\lambda_2), \tag{E.6}$$

where we kept the ratio $\xi_1/\xi_2$ to simplify the formula. This expression for $\epsilon_2'(\lambda_2)$ has to be complemented with expressions for $n_2$ and $e_2$. Together with a chemical potential of the second tube (which is a free parameter) they form a closed system that has to be solved.

### Conserved charges

We turn now our attention to conserved charges. As discussed in the main text, the relevant equation is

$$\frac{(f_1')_{\mathrm{dr},1}(\lambda_1)}{\omega_1'(\lambda_1)} - \frac{(f_2')_{\mathrm{dr},2}(\lambda_2)}{\omega_2'(\lambda_2)} = 0, \qquad v_1(\lambda_1) = v_2(\lambda_2). \tag{E.7}$$

This equation has the same structure as the stationary state equation. Therefore, we can immediately write down the solution

$$(f_2')_{\mathrm{dr},2}(\lambda_2) = \frac{\xi_2}{\xi_1}(f_1')_{\mathrm{dr},1}\left(\frac{\xi_1}{\xi_2}\lambda_2\right) + 2\left(\frac{e_1}{c_1^3} - \frac{e_2}{c_2^3} + \left(\frac{n_1}{c_1^3} - \frac{n_2}{c_2^3}\right)\lambda_2^2\right)\left(f_1'(\lambda_2) - \lambda_2 f_1''(\lambda_2)\right)$$
$$- 4\left(\frac{e_1}{c_1^3} - \frac{e_2}{c_2^3}\right)\lambda_2 f_1''(\lambda_2) + \mathcal{O}\left(1/c_i^4\right). \tag{E.8}$$

Note that in writing the terms of order $1/c_i^3$ we have neglected the dressing as its effect for those terms is of order $1/c_i^4$. Unlike for the stationary state, this equation has to be fulfilled at every time. However while the state of the system evolves so do the densities $\rho_{\mathrm{p},i}(\lambda)$ and hence the dressings change. Therefore, it is not clear whether we can find time-independent functions $f_i(\lambda)$ such that this equation holds. To verify this as a first step we rewrite it in terms of the bare quantities. To simplify the considerations we assume that $f_1(\lambda)$ is an even function. Then, according to Eqn. (D.14),

$$(f_1')_{\mathrm{dr},1}(\lambda) = f_1'(\lambda) + \frac{4\langle f_1'\rangle_1\lambda}{c_1^3} + \mathcal{O}\left(1/c_1^5\right), \qquad \langle f\rangle_i = \int \mathrm{d}\mu\,\rho_{\mathrm{p},i}(\mu)\mu f(\mu). \tag{E.9}$$

We note that the inverse operation is simply

$$f_1'(\lambda) = (f_1')_{\text{dr},1}(\lambda) - \frac{4\langle (f_1')_{\text{dr},1}\rangle_1 \lambda}{c_1^3} + \mathcal{O}\left(1/c_1^5\right). \tag{E.10}$$

We can now express $(f_2')_{\text{dr},2}$ in terms of bare $f_1'$. The result is

$$(f_2')_{\text{dr},2}(\lambda_2) = \frac{\xi_2}{\xi_1} f_1'\left(\frac{\xi_1}{\xi_2}\lambda_2\right) + \frac{4\lambda_2\langle f_1'\rangle_1}{c_1^3} - 4\left(\frac{e_1}{c_1^3} - \frac{e_2}{c_2^3}\right)\lambda_2 f_1''(\lambda_2)$$
$$+ 2\left(\frac{e_1}{c_1^3} - \frac{e_2}{c_2^3} + \left(\frac{n_1}{c_1^3} - \frac{n_2}{c_2^3}\right)\lambda_2^2\right)\left(f_1'(\lambda_2) - \lambda_2 f_1''(\lambda_2)\right). \tag{E.11}$$

We observe that $(f_2')_{\text{dr},2}$ is an odd function of $\lambda_2$. Finally we need to undress it to find bare $f_2'$. To achieve this we use the inverse operation (E.10)

$$f_2'(\lambda_2) = \frac{\xi_2}{\xi_1} f_1'\left(\frac{\xi_1}{\xi_2}\lambda_2\right) + 4\lambda_2\left(\frac{\langle f_1'\rangle_1}{c_1^3} - \frac{\langle f_1'\rangle_2}{c_2^3}\right) - 4\left(\frac{e_1}{c_1^3} - \frac{e_2}{c_2^3}\right)\lambda_2 f_1''(\lambda_2)$$
$$+ 2\left(\frac{e_1}{c_1^3} - \frac{e_2}{c_2^3} + \left(\frac{n_1}{c_1^3} - \frac{n_2}{c_2^3}\right)\lambda_2^2\right)\left(f_1'(\lambda_2) - \lambda_2 f_1''(\lambda_2)\right). \tag{E.12}$$

We can check whether this formula works for the total energy, which is always a conserved charge. Choosing $f_1(\lambda) = \lambda^2$ we observe that the second line vanishes. In the same time $\langle f_1'\rangle_1 = 2e_1$ and $\langle f_1'\rangle_2 = 2e_2$. This makes the second and the third term of the first line cancel out. We correctly find $f_2'(\lambda_2) = 2\lambda_2$.

For this equation to determine a conserved charge, the right hand side must be independent of time when $\rho_{\text{p},i}$ evolves according to the Boltzmann equation. The densities $n_i$ are conserved and therefore the time dependence enters only at order $1/c_i^3$ through $e_i$ and $\langle f_1'\rangle_i$. The time dependent part is

$$4\lambda_2\left(\frac{\langle f_1'\rangle_1}{c_1^3} - \frac{\langle f_1'\rangle_2}{c_2^3}\right) + 2\left(\frac{e_1}{c_1^3} - \frac{e_2}{c_2^3}\right)\left(f_1'(\lambda_2) - 3\lambda_2 f_1''(\lambda_2)\right). \tag{E.13}$$

It is a function of $\lambda_2$ and $t$. The time dependence enters through the averages $\langle \cdot \rangle_i$ and through the $e_i$ which are not conserved individually (only the total energy $e_1 + e_2$ is conserved). For this combination to vanish there are two options. First, $f_i(\lambda)$ are constant functions - this corresponds to the total density which indeed is conserved. Second, $f_1'(\lambda_2) - 3\lambda_2 f_1''(\lambda_2)$ must be a linear function of $\lambda_2$. This leads us again to the total energy. There are no other solutions and we conclude that beyond order $1/c^2$ it is not possible to find solutions to (E.7).

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
