# Peer review of "Prethermalization in coupled one-dimensional quantum gases"

_SciPost Physics, doi:SciPost Phys. 17, 007 (2024)_

## Round 1 · Referee Report · Anonymous (Referee 1) · 2023-6-15

Report

The authors study a system of two one-dimensional quantum bosonic gases, coupled through a density-density non-local interaction term. Each one-dimensional system is described by an integrable Lieb-Liniger model. However, the interaction between the two systems break integrability. The authors are interested in the late-time regime, when the system is initialized in a non-equilibrium state. While at infinite-times one expects thermal behavior to emerge, the authors study the case of weak integrability breaking and aim at characterizing the pre-thermal regime emerging at large but finite times.

This is a fully theoretical work, using a hydrodynamic framework based on the Boltzmann collision integral approach. Overall, I think the draft is very well written and proceeds in a very logical way. I could follow all the sections, and I believe the style of the presentation makes the material superficially accessible even by non-experts.

Having said this, I am a bit hesitant to recommend publication in Scipost Physics, as I believe Scipost Physics Core would be more appropriate. The main reason is that in this work there is no independent numerical result supporting the analytical predictions (by "independent" I mean some ab-initio numerical simulation, using for instance tensor-network techniques). Of course, I understand that producing numerics for this model might be very challenging or even impossible. Still, without an independent numerical test it is difficult to quantitatively estimate the impact of the approximations made in the hydrodynamic setting. Accordingly, in my opinion the main value of this work relies in its technical/mathematical aspects. They are certainly impressive, but perhaps of interest for a more specialized audience.

I also have a curiosity: it seems to me that the model is integrable in the limit opposite to that studied by the authors, namely when A(x)=\delta(x) (after regularization). In this case, it seems to me the model can be mapped onto a Yang-Gaudin Hamiltonian. Is this true? Do the authors expect that this observation can be used to study the dynamics in the opposite limit of short-range interactions?

In conclusions, I would recommend publication of the draft in Scipost Physics Core.
  • validity: -
  • significance: -
  • originality: -
  • clarity: -
  • formatting: -
  • grammar: -

Author:  Maciej Łebek  on 2024-03-15  [id 4367]

(in reply to Report 1 on 2023-06-15)

Dear Referee,

Thank you for feedback and comments on our manuscript. In the following, we would like to address the specific points raised in your report.

1) This is a fully theoretical work, using a hydrodynamic framework based on the Boltzmann collision integral approach.

In our approach, we do not rely on hydrodynamic approximation. The system is homogeneous and the Boltzmann equation can be systematically derived starting from the Fermi's Golden Rule (FGR), which is the lowest order of quantum time-dependent perturbation theory. Thus, it is different from the phenomenological hydrodynamic approach based on the gradient expansion and the existence of conserved charges. We use the standard FGR method (widely used in atomic physics and optics) to a situation of interacting many-body system.

2) The main reason is that in this work there is no independent numerical result supporting the analytical predictions (by "independent" I mean some ab-initio numerical simulation, using for instance tensor-network techniques). Of course, I understand that producing numerics for this model might be very challenging or even impossible. Still, without an independent numerical test it is difficult to quantitatively estimate the impact of the approximations made in the hydrodynamic setting.

Indeed, a reliable numerical check for such systems is a very difficult task. Homogeneous, non-equilibrium initial states that we consider are hard to prepare in tensor-network simulations especially that we are considering a continuum system, which brings additional challenges from the perspective of numerical simulations. Instead, our calculations are based on standard time-dependent perturbation theory truncated at the level of FGR. This is a systematic and established method, and there are no reasons, why it should break down, assuming that inter-tube interactions are sufficiently weak.

Furthermore, in a recent preprint co-authored by one of us (Ref. 51 in the manuscript), a similar prethermalization phenomena was observed in a fully quantum dynamics of the XXZ spin chains forming a ladder. This additionally secures that the results of our work are not peculiar to the Lieb-Liniger model and are not an artefact of the approximations that we perform.

3) Accordingly, in my opinion the main value of this work relies in its technical, mathematical aspects. They are certainly impressive, but perhaps of interest for a more specialized audience.

Apart from technical aspects involving properties of the Boltzmann collision integral, we emphasize that our work identifies a concrete and interesting physical phenomenon, i.e. existence of prethermalization plateaux in the relaxation dynamics. As was pointed out by the Second Referee, the setup of long-range coupled Lieb-Liniger gases is an important problem from the theoretical point of view. What is more, it was also studied in recent ultracold gases experiments (see Refs. 19 and 20 in our manuscript).

4) I also have a curiosity: it seems to me that the model is integrable in the limit opposite to that studied by the authors, namely when $A(x)=\delta(x)$ (after regularization). In this case, it seems to me the model can be mapped onto a Yang-Gaudin Hamiltonian. Is this true? Do the authors expect that this observation can be used to study the dynamics in the opposite limit of short-range interactions?

This is true and very interesting observation. This situation was studied in the spin chain ladder in the aforementioned preprint [51]. The results also show the prethermalization plateau. We also note that the Yang-Gaudin model requires inter- and intra-tube interactions to be identical. This makes it less relevant for the experiments with cold atoms.

5) In conclusions, I would recommend publication of the draft in Scipost Physics Core.

After the feedback from the Referees, we still believe that our manuscript deserves publication in SciPost Physics. Both the setup and uncovered phenomenon are interesting for general audience. Most of the technical aspects of the work are related to identification of the conserved charges. On the other hand, the most important part (existence of a prethermalization plateau) only involves the Boltzmann equation formalism and the Thermodynamic Bethe Ansatz. The latter is a standard tool, as illustrated by its ubiquity in generalized hydrodynamical computations, to study thermodynamics and dynamics of low-dimensional many-body systems.

---

## Round 1 · Referee Report · Anonymous (Referee 2) · 2023-6-25

Strengths

1- good numerical results to support the proposed prethermalisation 2- nicely written 3- important subject

Weaknesses

1- explanations in terms of extra non-trivial conserved charge unclear or incorrect

Report

In this work, the authors study the evolution of two Lieb-Liniger models coupled by density-density coupling in the limit of small characteristic coupling momentum and with different states / tubes being coupled. They use the Boltzmann equation formalism. In this limit, only certain processes (called (1,1)-processes) are involved, which, it is claimed, do not lead to thermalisation. Convincing numerical results are provided which show that this is indeed the case. Thus, there is a pre-thermalisation plateau. The authors also provide an explanation in terms of additional, non-trivial conserved quantities that are supposedly admitted by the coupled model.

I think the overall results are good and interesting. The subject of thermalisation / prethermalisation due to integrability breaking is certainly of high interest currently, and the setup of coupled Lieb-Liniger model is very relevant. Admittedly this is a small extension / adaptation of some previous works, however it does unveil an interesting phenomenon.

But I do not understand the explanations in terms of the non-trivial conserved quantities. It seems to me that the explanations are incorrect, and that the authors do not actually explain the phenomenon in terms of such non-trivial local conserved quantities. I agree that there are objects in the Boltzmann equations that constrain the (1,1)-dynamics, but I don't think these have been correctly identified with extra local conserved quantities (or it may be that I have misunderstood something).

Thus, I would accept the paper for publication only after this has been clarified or re-written in order to be correct.

Precise comments:

After eq 16 say that "Q_0 is defined below" or the like for more readability

After eq 33: perhaps also say that we need the operation of dressing to be invertible to conclude that we must have vanishing of Q_0.

Just before eq 34: say that this is indeed a necessary condition as all other factors in eq 29 are strictly positive (on the set eqs 36,37).

Page 12 "energy momentum-energy conservation laws"

Equation 58 is valid to what order in 1/c? First order only? All orders? Please clarify.

Solution to eq 59: it is stronger than that, as what we want is to find $f_1$ and $f_2$ that are *independent of the state* (as in the value of conserved quantities on the state, eq 49, the functions $f_i(\lambda)$ do not depend on the state). The state-independent $f_i$ must satisfy eq 59 for all states. It is very different from eq 56, where $\epsilon_i$ are determined by (characterise) the state. The problem appears to be very different. The special cases consider, where dressing disappear, are ok, but the full set of solutions will be different it seems.

Pages 14-16: The language and discussion of conserved charges and GGEs is confusing and, it seems, the discussion is not entirely correct.

Eq 62 is not a conserved charge, it is the average value of the conserved charge in the state characterised by $\rho_{p,i}(\lambda)$ (GGE of the unperturbed system). Eq 64 is unclear: what is the time $T$? What is the specific dynamics? It in fact does not make sense, as the average value of the conserved charge should only depend on the state, and should not have explicit time dependence (there is no time, there is just a state). Do the authors want to take $T\to\infty$? Is the idea that $\bar I_{i,j}$ is then a function of the state, specifically the time-averaged value of $I_{i,j}(t)$ starting form that state? Please clarify this. If so, then indeed $\mathcal I_j$ is a function of the state only, and this would make more sense.

However, even then, the result would be a very nonlinear function of $\rho_{p,i}(\lambda)$. Then, of course, there is little guarantee that this is the average of a local operator.

On this point, the authors said that locality of $\mathcal I_j$ would be clear. But then, in order for this to be so, one should interpret eq 62 more "literally" as an operator equation, not just an equation between average. Just an equation between averages in GGEs (of the unperturbed system) would not guarantee that $\mathcal I_j$ is local as an operator; there are many nonlocal operators that have the same expectation values as local operators in all GGEs. Then this should be made clear, perhaps with hatted symbols, and in this case $\hat I_{ij}$ is not given by eq 63 (which is just its average).

But then, this would be a problem, as the definition of the operator $\mathcal I_j$ would involve explicitly not only the charges $I_{i,j}$, but also denominators that depend explicitly on the state. We cannot define intrinsically a charge with coefficients that depend on the state (we can take linear combinations of conserved quantities with state-dependent coefficients, e.g. in subtracting the average times the identity operator, but then this is not intrinsic). Thus, eq 62 as an operator equation does not make sense.

I conclude (if the limit $T\to\infty$ is implied) that the authors define rather complicated state-dependent functions, eq 62, which they would like to interpret as averages of local conserved quantities in the state, but there is no indication that these correspond to local conserved quantities.

Other questions are: how does one minimise the fluctuations (sentence after eq 65)? Why does on do so? How is the condition of non-zero norm of the operator translate to eq 66? Given that only averages of charges in states were defined, how does one get to an operator norm condition?

I agree nevertheless that the (1,1)-induced dynamics may be constrained, and that functions of the state that are invariant under this dynamics indeed constrain it and may stop it from thermalising. But I am not convinced that the discussion here explains this in terms of any nontrivial local conserved quantities of the low-$k_r$ perturbation.

Eq 68: I don't understand this. Naively this seems to be try from eq 67 with eq 62. However, in eq 62 each denominator seems to be a function of both $\rho_{p,1}$ and $\rho_{p,2}$, because the time evolution involves coupling between them (unless I am missing something); and also perhaps the coefficients because of the minimisation of fluctuations.

Then, I am rather lost on the rest of that section. In particular, in eq 70 there is $f_{ij}(\lambda)$, the one-particle contribution of nontrivial conserved quantities. But if the source term of the pseudo energy has this form, then the average of the nontrivial conserved quantities should be linear in $\rho_{p,i}$. But I concluded above that this couldn't be the case from the definition eq 62. So this seems to be inconsistent.

Requested changes

see above

  • validity: good
  • significance: good
  • originality: ok
  • clarity: ok
  • formatting: perfect
  • grammar: excellent

Author:  Maciej Łebek  on 2024-03-15  [id 4368]

(in reply to Report 2 on 2023-06-25)

Dear Referee,

thank you for feedback and comments on our manuscript. In the following, we would like to address the specific raised points in your report.

1) After eq 16 say that "$Q_0$ is defined below" or the like for more readability

Thank you for this comment. We have modified the text below eq. 16.

2) After eq 33: perhaps also say that we need the operation of dressing to be invertible to conclude that we must have vanishing of $Q_0$.

We have added the comment.

3) Just before eq 34: say that this is indeed a necessary condition as all other factors in eq 29 are strictly positive (on the set eqs 36,37)

We have expanded the discussion around eq. (34) by stating explicitly that the bare integral can vanish as a whole or because the integrand is zero. In the latter case, because the other factors are strictly positive, this leads to a necessary condition of eq. 29.

4) Page 12 "energy momentum-energy conservation laws"

Corrected.

5) Equation 58 is valid to what order in $1/c$? First order only? All orders? Please clarify.

It is valid up to and including $1/c^2$. See eq. (62) of the new version.

6) Solution to eq 59 (now eq. 68): it is stronger than that, as what we want is to find $f_1$ and $f_2$ that are independent of the state (as in the value of conserved quantities on the state, eq 49, the functions $f_i(\lambda)$ do not depend on the state). The state-independent $f_i$ must satisfy eq 59 for all states. It is very different from eq 56, where $\epsilon_i(\lambda)$ are determined by (characterise) the state. The problem appears to be very different. The special cases consider, where dressing disappear, are ok, but the full set of solutions will be different it seems.

We agree with the referee and we emphasise now the difference between the equations for the stationary state and for the charges in the text. Furthermore, we now explicitly show that the equation for the conserved charges can be still solved in the strongly interacting case that we dub the deformed Tonks-Girardeau gas. The deformed Tonks-Girardea gas includes $1/c$ corrections to the Tonks-Girardeau gas up to and including order $1/c^2$. In this case we find that $f_i$ depend on the state only through the total density of particles in each tube - a quantity that is an invariant of the dynamics. Furthermore, in Appendix E, we show that it is impossible to find such conserved charges beyond the deformed Tonks-Girardeau case.

7) Eq 62 is not a conserved charge, it is the average value of the conserved charge in the state characterised by $\rho_{p,i}(\lambda)$ (GGE of the unperturbed system). Eq 64 is unclear: what is the time T? What is the specific dynamics? It in fact does not make sense, as the average value of the conserved charge should only depend on the state, and should not have explicit time dependence (there is no time, there is just a state). Do the authors want to take $T\rightarrow \infty$? Is the idea that $I_{i,j}$ is then a function of the state, specifically the time-averaged value of $I_{i,j}(t)$ starting form that state? Please clarify this. If so, then indeed $I_j$ is a function of the state only, and this would make more sense.

Upon further studies of our numerical construction of the charges, we have decided to abandon it for now. We have found that observing the minimisation of the fluctuations is not a sufficiently sensitive condition to define a conserved charge. This is something to which we plan to return, but for the present work, we have decided to shift our attention to the analytical results available at large $c$.

We would like to thank the referee for the comments on the numerical construction and replying to the referee comment, indeed we meant to take the $T\rightarrow \infty$ limit.

8) On this point, the authors said that locality of $I_j$ would be clear. But then, in order for this to be so, one should interpret eq 62 more "literally" as an operator equation, not just an equation between average. Just an equation between averages in GGEs (of the unperturbed system) would not guarantee that $I_j$ is local as an operator; there are many nonlocal operators that have the same expectation values as local operators in all GGEs. Then this should be made clear, perhaps with hatted symbols, and in this case $I_{i,j}$ is not given by eq 63 (which is just its average). But then, this would be a problem, as the definition of the operator $I_j$ would involve explicitly not only the charges $I_{i,j}$, but also denominators that depend explicitly on the state. We cannot define intrinsically a charge with coefficients that depend on the state (we can take linear combinations of conserved quantities with state-dependent coefficients, e.g. in subtracting the average times the identity operator, but then this is not intrinsic). Thus, eq 62 as an operator equation does not make sense.

The construction of the charges for the deformed Tonks-Girardaeu case allows us to address this point directly. The charges have then coefficients that depend on the state only through gas densities which are conserved by the evolution. Thus, working in the Hilbert space with fixed number of particles, these charges correspond to linear operators. On the other hand, going beyond the deformed Tonks-Girardeau gas, as we now understand, invalidates the whole construction because we are unable to solve for functions $f_{i,j}$ and define $I_{i,j}$. We then agree with the referee that the old eq. 62 would not make sense at the operatorial level for generic values of $c$. However again, limiting ourselves to the large $c$ regime of the deformed Tonks-Girardeau gas, avoids this issue.

9) how does one minimise the fluctuations (sentence after eq 65)? Why does on do so? How is the condition of non-zero norm of the operator translate to eq 66? Given that only averages of charges in states were defined, how does one get to an operator norm condition?

The minimization was achieved by solving a linear problem for the coefficients and we were using it as a criteria to validate that the proposed quantity is a conserved charge. We were working only at the level of averages not trying to relate directly the resulting objects with operators conserved at the quantum mechanical level. We, in any case, have now abandoned the numerical construction as we found it was not sufficiently sensitive to indicate whether or not a conserved quantity was actually present.

10) Eq 68: I don't understand this. Naively this seems to be try from eq 67 with eq 62. However, in eq 62 each denominator seems to be a function of both $\rho_{p,1}$ and $\rho_{p,2}$, because the time evolution involves coupling between them (unless I am missing something); and also perhaps the coefficients because of the minimisation of fluctuations.

In the present version of the manuscript we have limited the GGE construction to the deformed Tonks-Girardeau gas where the expressions for the conserved charges are explicit. In the Hilbert space with fixed number of particles they are linear operators. This allow us to construct the GGE for a coupled system in an analogous way for a single Lieb-Liniger gas. The main difference occuring for the coupled system is a renormalization of the the chemical potential that occurs because the conserved charges have coefficients depending on the density of the particles.

11) Then, I am rather lost on the rest of that section. In particular, in eq 70 there is $f_{i,j}(\lambda)$, the one-particle contribution of nontrivial conserved quantities. But if the source term of the pseudo energy has this form, then the average of the nontrivial conserved quantities should be linear in $\rho_{p,j}$. But I concluded above that this couldn't be the case from the definition eq 62. So this seems to be inconsistent.

This tension is now resolved by limiting the construction of the GGE to the deformed Tonks-Girardeau gas. The conserved charges of the coupled system are given then by the usual ultra-local conserved charges multiplied by a particle number dependent prefactors. Working in the Hilbert space of fixed particle number, their eigenvalues on a multi-particle state are linear in $\rho_{p, j}$.

---

## Round 2 · Referee Report · Anonymous (Referee 2) · 2024-4-16

Report

I am happy with the changes that the authors have done. I think all the points I raised have been addressed. I think it can be published now.

I think the physics extracted is interesting. Certainly this is a technical calculation and specific to the Lieb Liniger model, but as the authors argue the physics may be more universal. I am not sure this is Scipost Physics, as it is not clear it satisfies the criteria as expressed in https://scipost.org/SciPostPhys/about, perhaps more Scipost Physics Core, but this is certainly a strong paper.

Recommendation

Publish (meets expectations and criteria for this Journal)

  • validity: -
  • significance: -
  • originality: -
  • clarity: -
  • formatting: -
  • grammar: -

Author:  Maciej Łebek  on 2024-05-07  [id 4477]

(in reply to Report 1 on 2024-04-16)

Dear Referee,

Thank you for your recommendation and for reading the new version of our manuscript. We are convinced that our work meets the acceptance criteria for SciPost Physics. Let us provide a rationale for choosing this journal.

Firstly, let us address the issue of the universality of the observed prethermalization phenomena argued for in the Report. In the recent paper, Ref. 51 (published as Phys. Rev. B 109, L161109) it is shown that similar physics emerges in weakly perturbed integrable spin chains. Importantly, in that work the full quantum dynamics was simulated thus proving that prethermalization plateau is not an effect of approximations inherent to the collision integral approach. These results, together with those presented in our manuscript, show that weakly perturbed integrable systems with direct relevance to experimental condensed matter and cold atomic system can exhibit prethermalization. This, we believe, will motivate further studies in this direction. Our work in particular has direct experimental consequences for thermalization or lack thereof in the one dimensional cold atomic gases studied by Ben Lev’s group at Stanford.

Secondly, we would like to mention the recent preprint (arXiv:2404.14292), in which we show that collision integrals together with generalized hydrodynamics lead to Navier-Stokes hydrodynamics in inhomogenous settings. In this new work we provide explicit computations of the transport coefficients which would not be possible without the studies of the collision integral performed in the work under the consideration.

Thirdly and finally, the research presented in this manuscript has a natural extension. In the project, which we are finishing now, we consider long-range integrability breaking couplings within one tube (as opposed to the two coupled tubes studied in this work). This is a new mechanism of integrability breaking and the detailed analysis of the collision integral found in the present work was crucial to understanding how to develop the methodology for this new physical situation.

All of this together, in our opinion, shows that the present work both addresses a relevant physical phenomena, prethermalization, and at the same time serves as a source of inspiration and methods in the wider context of the dynamics of weakly perturbed integrable models. As such it is of interest to the general community of readers at SciPost Physics.

Kind regards,
Authors

---

## Round 2 · Author Response

Dear Editor,

We are sending a new version of the manuscript which has been thoroughly revised following the referees' reports and our further work on the subject. The main changes are:

  1. We have included a detailed analysis of the dynamics when both tubes are in the Tonks-Girardeau regime or a slight deviation from it, dubbed deformed Tonks-Girardeau gas. The analysis is in Appendix C and D and main findings are explained in the main text.

  2. Specifically for the deformed Tonks-Girardeau gas, we prove the existence of conserved charges which are linear combination of the charges present in the uncoupled system.

  3. At the same time, we show that such construction of the charges does not extend to the full Lieb-Liniger model. This is shown in Appendix E.

  4. Given our analytic control over the charges for the deformed Tonks-Girardeau gas, we construct the generalized Gibbs ensemble. The construction shares many similarities with GGE of a single tube, but also has a new quality. As the conserved charges depend explicitly on the density of the gas in the tubes, this causes a renormalization of the chemical potentials.

  5. Upon further considerations we have decided to abandon the numerical construction of the charges found in the first version of our manuscript. We have found that the numerical test that we were using to show that the charges were conserved was not sufficiently discriminatory. While we still believe that such numerical constructions are useful, presently we decided to shift our attention to analytic results.

We have also decided to rearrange the structure of our manuscript. The numerical results that constituted a separate section are now a part of Section 4, whereas a discussion of conserved charges for (1,1) processes, previously in Section 4.2, is now in Section 5.1. The GGE construction is in Section 5.2.

---

## Round 2 · List of Changes

Below we list the main changes:

  1. clarified the expression for the scattering integral between Eqns. (16) and (17).
  2. clarified that the dressing operation is invertible in explanation of the stationary states between Eqns. (33) and (34).
  3. rephrased the text to indicate that the condition for the stationarity given in eq. (35) is sufficient but not necessary.
  4. we have extended the discussion on conserved charges in Section 3.3, below Eq. (48)
  5. we have clarified that in the Tonks-Girardeau limit the prethermal state exists beyond the small momentum limit.
  6. we have clarified that the condition for stationary state in Eq. (62) is valid up to corrections of order (1/c_i)^3.
  7. Section 4.2 with construction of charges in the small momentum limit has been moved to Section 5.1
  8. we have removed the numerical construction of the charges from Section 4.3 and the corresponding figures.
  9. we have rewritten Section on the GGE by limiting it to the deformed Tonks-Girardeau gas. At the beginning of this Section we introduce now conserved charges present in that case.
  10. we have added a new figure 7 to show that the first conserved charge beyond the energy in the deformed Tonks-Girardeau gas is indeed conserved. We also show that the construction of this charge does not extend to smaller values of intra-tube interactions.
  11. we have added a new figure 9 to show that the approach to a stationary state can be understood as equilibration of the generalized chemical potentials, thus confirming the predictions of the GGE.
  12. Section 6 with numerical results is now a subsection 4.3 of Section 4 on the (1,1) dynamics in the small momentum limit.
  13. New Appendix C in which we describe in details the scattering integral, stationary states and conserved charges in the Tonks-Girardeau gas
  14. New Appendix D in which we present analogous results for the deformed Tonks-Girardeau gas, that is including corrections of order (1/c_i)^2
  15. The old Appendix C is now Appendix E. In this Appendix we show how including (1/c_i)^3 corrections prevents a time-independent solution to equation (68) defining the conserved charges.
  16. We have also adjusted introduction and summary to be in line with our revision of the manuscript.

---

## Editorial Decision

published